# Unique neural coding of crucial versus irrelevant plant odors in a hawkmoth

**Sonja Bisch-Knaden[1]\*, Michelle A Rafter[2], Markus Knaden[1], Bill S Hansson[1]**

[1]Max-Planck-Institute for Chemical Ecology, Department of Evolutionary Neuroethology, Jena, Germany; [2]CSIRO Health and Biosecurity, Dutton Park Queensland, Australia

**Abstract** The sense of smell is pivotal for nocturnal moths to locate feeding and oviposition sites. However, these crucial resources are often rare and their bouquets are intermingled with volatiles emanating from surrounding 'background' plants. Here, we asked if the olfactory system of female hawkmoths, *Manduca sexta*, could differentiate between crucial and background cues. To answer this question, we collected nocturnal headspaces of numerous plants in a natural habitat of *M. sexta*. We analyzed the chemical composition of these headspaces and used them as stimuli in physiological experiments at the antenna and in the brain. The intense odors of floral nectar sources evoked strong responses in virgin and mated female moths, most likely enabling the localization of profitable flowers at a distance. Bouquets of larval host plants and most background plants, in contrast, were subtle, thus potentially complicating host identification. However, despite being subtle, antennal responses and brain activation patterns evoked by the smell of larval host plants were clearly different from those evoked by other plants. Interestingly, this difference was even more pronounced in the antennal lobe of mated females, revealing a status-dependent tuning of their olfactory system towards oviposition sites. Our study suggests that female moths possess unique neural coding strategies to find not only conspicuous floral cues but also inconspicuous bouquets of larval host plants within a complex olfactory landscape.

**\*For correspondence:**
sbisch-knaden@ice.mpg.de

**Competing interest:** The authors declare that no competing interests exist.

## Editor's evaluation

This article is of particular interest to researchers in the fields of neuroecology of insect olfaction and of insect–plant interactions in general. The authors investigate the olfactory signals that guide the specialist hawkmoth *Manduca sexta* towards plants that are used for oviposition and for nectar-feeding in a natural setting. How insects distinguish useful information from irrelevant information is an important question. The authors use elegant chemical ecology techniques and recordings of neuronal activity to ask how female moths (*M. sexta*) could discriminate co-occurring behaviorally relevant versus irrelevant plant and floral volatiles.

## Introduction

Nocturnal insects largely rely on their sense of smell to locate food sources and oviposition sites. However, preferred nectar sources or suitable host plants are often rare, and odors emitted by these essential plants are mixed with bouquets released by neighboring plants (*Bruce et al., 2005*). Plants that depend on nocturnal pollination, for example, by hawkmoths, advertise their nectar-providing flowers with bright colors, and most notably by a strong scent to attract their nectar-feeding pollinators (*Raguso et al., 2003a*; *Raguso et al., 2003b*). Evolution has thus formed a system that greatly facilitates the location of nectar sources by foraging insects. The situation is very different when gravid females search for a suitable host plant for oviposition. Vegetative parts of plants, probably to remain

cryptic to herbivores, emit only trace quantities of volatiles that might be difficult to identify against the olfactory background provided by other plants (*Turlings et al., 1995*). At the same time, leaf damage by insect herbivores leads to an increased emission of volatiles, sometimes attracting parasitoids and predators of these insects (*Pare and Tumlinson, 1999*). Herbivore-induced plant volatiles are often similar across plant species (*Mumm and Dicke, 2010*), and herbivores are omnipresent in natural environments. Together, these facts suggest that female moths searching for an oviposition site encounter either undamaged, olfactorily unremarkable plants or damaged plants with a more conspicuous volatile profile, components of which are shared across many non-host plants. In any case, gravid moths have to identify suitable host plants against a vast odor scenery provided by the background vegetation. For this purpose, insects might depend on taxon-specific volatiles released by host plants but not by surrounding plants. An example is the cabbage moth *Plutella xylostella* that uses host plant-specific isothiocyanates to locate cruciferous hosts (*Liu et al., 2020*). Most insects, however, seem to identify host plants by blends of ubiquitous components that are present in plant-specific ratios. This has been shown in experiments where small changes in the composition and ratios of crucial odor blends had a huge impact on the behavior of insects (*Cha et al., 2008*; *Kárpáti et al., 2013*; *Riffell et al., 2014*; *Visser and Avé, 1978*; *Webster et al., 2010*). Furthermore, the chemical composition of crucial blends and the capability of the insect's antenna to detect specific components of these blends have been studied in detail (*Conchou et al., 2017*; *Fraser et al., 2003*; *Tasin et al., 2010*). At the level of the first olfactory processing center in the insect brain, the antennal lobe, previous studies have investigated the temporal coding patterns evoked by floral bouquets. By impaling a restricted region of the antennal lobe with a multiunit probe, simultaneous recordings from a mixed population of local interneurons and projection neurons in this area were performed (*Riffell et al., 2009a*; *Riffell et al., 2009b*). Using the same approach, inhibitory interactions between these neurons could be studied in addition (*Lei et al., 2004*). However, this recording technique does not allow assigning functional significance to individual olfactory glomeruli, which are the morphological and functional subunits of the antennal lobe (*Gao et al., 2000*; *Hansson et al., 1992*). An analysis of the spatial coding patterns, however, is possible via functional calcium imaging. Although this technique does not inform about temporal coding patterns or inhibitory interactions, it was used in different insect species to provide a detailed insight into the spatial representation of natural odor blends across the glomerular array (*Burger et al., 2021*; *Lahondère et al., 2020*; *Saveer et al., 2012*; *Schubert et al., 2014*; *Zhao et al., 2020*). Usually, only odor blends that are known to be essential in the ecology of the insect were tested as the aim of those studies was to reveal how crucial blends are coded in the brain. However, it remains unclear how the olfactory systems of insects can differentiate between crucial and irrelevant blends, that is, how peripheral detection and central representation allow the identification of food sources and oviposition sites within a complex olfactory environment.

In our study, we collected headspaces of focal plants and background vegetation in the habitat of the tobacco hawkmoth *Manduca sexta* in Southern Arizona. Volatiles of hawkmoth-visited plants, like of most vegetation, differ between day and night both regarding floral (*Hoballah et al., 2005*; *Raguso et al., 2003b*) and leaf emissions (*De Moraes et al., 2001*). We collected plant headspace only during the night as we were interested in how the nocturnal *M. sexta* would detect and process these olfactory cues.

The primary nectar sources for *M. sexta* in the Southwestern United States are flowers of *Agave palmeri* and *Datura wrightii*. Volatile emissions of these two species are strong but very different from each other (*Raguso, 2004*; *Raguso et al., 2003a*; *Riffell et al., 2008*), and their pollen together account for 90% of the pollen load on the proboscis of *M. sexta* (*Alarcón et al., 2008*), a measurement that can be used as a proxy for flower visitation. Presence of pollen from *Mirabilis longiflora* and *Mimosa dysocarpa* on the moth's proboscis and nighttime observations reveal that these plants are additional secondary nectar sources in the same habitat (*Alarcón et al., 2008*; *Grant and Grant, 1983*). *Datura,* in addition to being a valuable nectar source for *M. sexta,* is one of its two larval host plants in the area. *Datura* plants thus have to interact with an insect that is at the same time an important pollinator and a damaging herbivore (*Bronstein et al., 2009*), enabling the moth to find an oviposition site by navigating towards the scent of nectar-providing flowers emitted by the same plant (*Reisenman et al., 2010*). The other local host plant of *M. sexta* larvae is *Proboscidea* spp., the only known host belonging to a non-solanaceous family (*Mechaber and Hildebrand, 2000*). Flowers of *Proboscidea,* however, are not visited by foraging hawkmoths, that is, *Proboscidea* plants are suffering

**Table 1.** Headspace collections from plants at the Santa Rita Experimental Range in Arizona (US).

| Plant species (plant family), common name | Type of sample | Nectar source for adult *M. sexta* | Host plant for *M. sexta* larvae | Larval host plant for sympatric hawkmoths | Nocturnal pollination |
|---|---|---|---|---|---|
| *Agave palmeri* (Asparagaceae), Palmer's century plant | Flower | X | – | – | X |
| *Datura wrightii* (Solanaceae), Sacred datura | Flower    Branch | X | X | X* | X |
| *Mimosa dysocarpa* (Fabaceae), Velvetpod | Flowering branch | X | – | – | X |
| *Mirabilis longiflora* (Nyctaginaceae), Sweet four o'clock | Flowering branch | X | – | – | X |
| *Proboscidea parviflora* (Martyniaceae), Devil's claw | Flowering plant | – | X | – | – |
| *Chilopsis linearis* (Bignoniaceae), Desert willow | Branch with seeds | – | – | X† | – |
| *Helianthus annuus* (Asteraceae), Common sunflower | Flowering plant | – | – | X‡ | X |
| *Vitis arizonica* (Vitaceae), Wild grape | Branch | – | – | X§ | – |
| *Amaranthus palmeri* (Amaranthaceae), Carelessweed | Flowering plant | – | – | – | – |
| *Argemone pleiacantha* (Papaveraceae), Prickly poppy | Flowering branch | – | – | – | – |
| *Baccharis salicifolia* (Asteraceae), Seepwillow | Branch with buds | – | – | – | – |
| *Gutierrezia sarothrae* (Asteraceae), Snakeweed | Flowering plant | – | – | – | – |
| *Poaceae* spp,, Grass | Tuft of grass | – | – | – | – |
| *Prosopis velutina* (Fabaceae), Velvet mesquite | Branch | – | – | – | – |
| *Quercus emoryi* (Fagaceae), Emory oak | Branch | – | – | – | – |
| *Senna hirsuta* v *glaberrima* (Fabaceae), Woolly Senna | Flowering plant | – | – | – | – |

*M. quinquemaculata.

†M. rustica, M. florestan.

‡M. muscosa.

§Eumorpha achemon.

from leaf consumption by *M. sexta* larvae but do not profit from pollination by ovipositing moths. Furthermore, we sampled odors from another 11 native, frequent plants in the direct neighborhood of *M. sexta*'s focal plants. These background plants have no documented relevance for *M. sexta*; they included flowering herbaceous plants, nonflowering woody shrubs or trees and tufts of grass. Three of the background plants, the desert willow *Chilopsis linearis*, the sunflower *Helianthus annuus*, and the wild grape *Vitis arizonica*, are larval hosts of other sympatric hawkmoth species (*Table 1*).

After collecting all nocturnal plant headspaces in situ in the field, we proceeded to analyze this comprehensive chemical database. We then used the plants' headspaces as stimuli in physiological experiments with female *M. sexta*. Specifically, we investigated which components of the volatile blends the moth's antenna can detect, and how the glomerular array of the antennal lobe is coding these complex odor bouquets. An insect's reaction to olfactory cues is known to be plastic in relation to its physiological condition and experience (*Gadenne et al., 2016*). The moths tested in our study were laboratory-reared on artificial diet, naïve to plant odors, not fed, and tested only once, as we were interested in the insects' innate neuronal responses. However, *M. sexta* has been demonstrated to differentially respond to plant odors depending on its mating status (*Mechaber et al., 2002*). Underlying this differential response is a state-dependent modulation of the olfactory system, which may take place at the level of the antenna, the brain, or at both levels (*Gadenne et al., 2016*; *Saveer*

*et al., 2012*). Therefore, we investigated the peripheral detection of plant headspace and the central representation of this olfactory information in both virgin and mated *M. sexta* females.

Our results revealed that the olfactory system of female moths responds strongly to odors related to nectar sources. Suitable oviposition substrates elicited much weaker but specific responses, a specificity that was most pronounced in gravid females. Evolution thus seems to have shaped an olfactory system that allows efficient feeding at all stages and that enables the mated female to pinpoint an optimal home for her offspring.

## Results

### Nocturnal emissions of plants in the habitat of *M. sexta*

We collected the nocturnal headspaces of 17 plant species at the Santa Rita Experimental Range, our study site in Southern Arizona (*Figure 1A*, *Table 1*). Headspace samples were analyzed chemically by gas chromatography coupled with mass spectrometry (GC-MS). We first evaluated the number of GC-peaks per sample as a proxy for the number of volatile compounds present. In 10 of the 17 plant samples, the number of emitted compounds was in the range of blank control collections (*Figure 1B*, gray area). The richest volatile bouquets, on the other hand, were emitted by the sunflower *Helianthus*, and by *M. sexta*'s nectar sources *Datura* flower, *Agave* flower, and *Mirabilis*. When we considered not only the number of GC-peaks but also their chemical identity, the same four bouquets revealed distinct chemical profiles. Headspaces of the remaining plants were statistically distinctive but largely overlapping due to low emission rates and shared volatiles, which were also present in the blank control samples (*Figure 1C*; one-way ANOSIM, $R = 0.67$, p<0.0001; Bray–Curtis similarity index).

### What does the moth detect?

So far, our analysis considered the chemistry of nocturnal plant emissions. However, *M. sexta* might still be able to detect plant volatiles occurring only in trace amounts but having a high biological significance, for example, to identify an appropriate oviposition site. Therefore, we performed GC-coupled electro-antennographic detection (GC-EAD) using the antennae of female *M. sexta* as biological detectors. This technique allows successive presentation of headspace compounds in naturally occurring concentrations to the moth antenna and, in parallel, recording of the pooled response of all antennal olfactory sensory neurons (*Figure 2A*). We first evaluated the number of EAD-active fractions in the effluent of the GC for each sample type (*Figure 2B*). With the exception of two background plants (*Argemone*, *Gutierrezia*), all plant bouquets contained EAD-active fractions. The nectar sources *Agave* flower and *Datura* flower emitted the highest number of compounds (on average, 20 and 17 active fractions, respectively), followed by the bouquets of host plants of sympatric hawkmoths (*Chilopsis*, *Helianthus*, *Vitis*) and a background tree (*Prosopis*) (11–14 active fractions). The two larval host plants of *M. sexta*, on the other hand, contained only 4–7 active compounds.

Across all headspaces, we found 77 EAD-active compounds (*Figure 2C*) and could tentatively identify 69 of them. These compounds mainly belonged to three chemical classes: terpenes, aliphatic esters, and aromatics. The most potent antennal stimulants (n = 16) elicited median EAD amplitudes > 1.0 mV. Eight of these strongly activating odors were aliphatic esters present exclusively in the bouquet of *Agave* flowers; three more odors were present in the headspace of nectar sources (*Agave* flower, *Datura* flower, and/or *Mirabilis*) but not in other sample types. The remaining strongly activating odors each occurred in at least five plant species from all sample types and included the most common volatiles in our collections: (Z)-3-hexenyl acetate (11 plants) and β-ocimene (10 plants). When we plotted the concentration of the most activating GC fractions versus the EAD amplitude they evoked, we found that α-copaene, (Z)-3-hexenyl acetate, and β-ocimene were the most active odors at concentrations below 5 ng in 12 hr of odor collection (*Figure 2D*).

The antenna of *M. sexta* was in addition reacting with a weaker response towards many more plant-released volatiles in a species-specific manner. Furthermore, two-thirds of all EAD-active GC fractions (51 out of 77) were restricted to one of the plant species (*Figure 2C*). Thus, beyond the impression received from the chemical analysis (*Figure 1C*), the moths' antennae seemed to be well suited to distinguish between plant bouquets even when they had low volatile concentrations and inconspicuous chemical profiles, like the two larval host plants of *M. sexta* and most background plants. The mating status of the moth had no impact on its detection capabilities at the level of the antenna

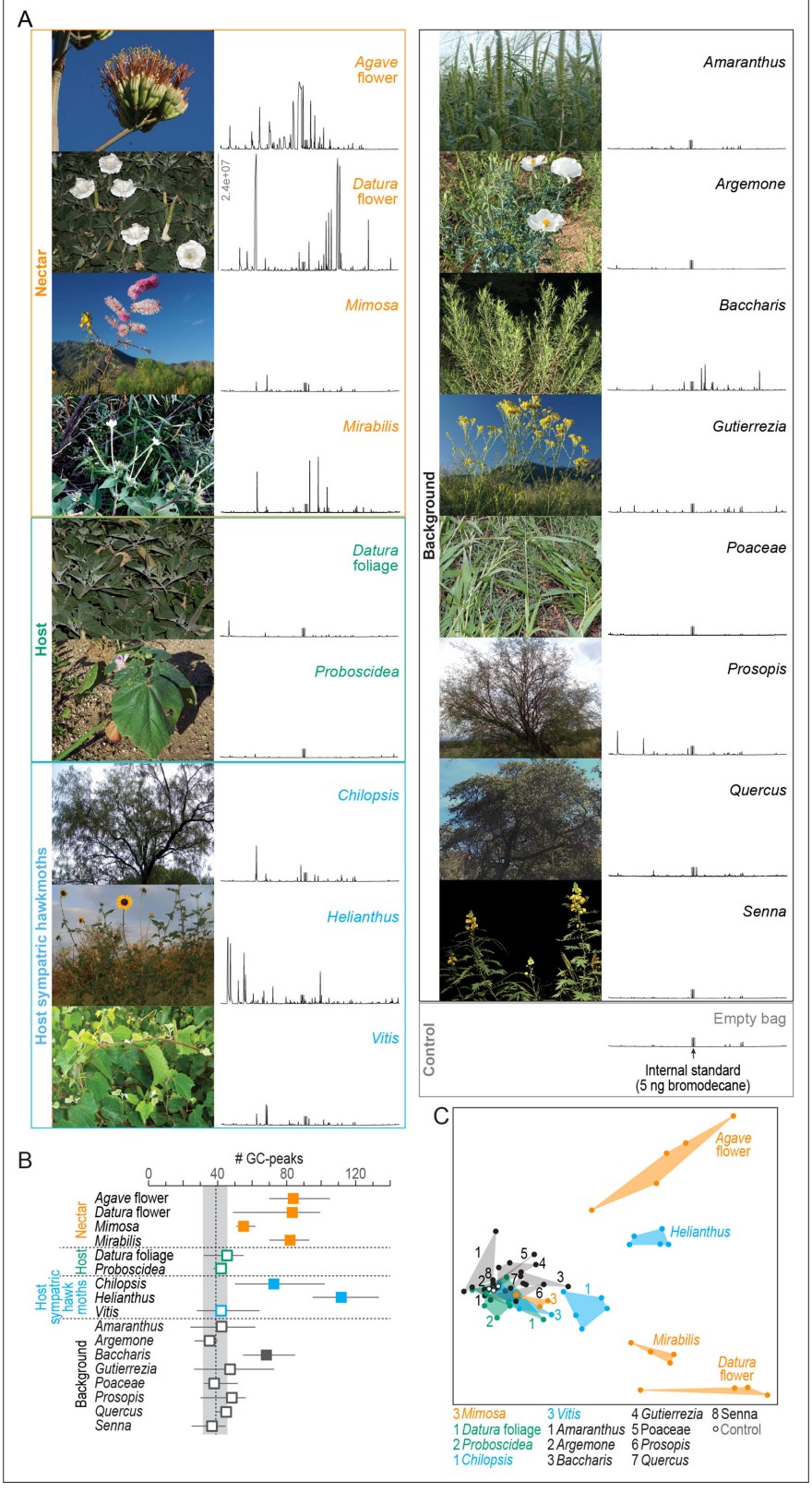

**Figure 1.** Chemical analysis of nocturnal headspaces collected from plants in the habitat of *M. sexta* in Southern Arizona. (**A**) Representative photographs (left) and chromatographs (right) of each headspace collection. x-axis of chromatographs, retention time; y-axis, abundance, same scale for all headspaces, maximum abundance indicated in *Datura* flower headspace; gray bar, internal standard (5 ng 1-bromodecane). (**B**) Number of GC-peaks. Squares,

*Figure 1 continued on next page*

*Figure 1 continued*
average values of 3–5 individual plant samples; whiskers, range; dotted line and gray area, average and range of control values obtained from nocturnal collections in the same habitat with empty bags (n = 2), and with unused filter material (n = 1); open squares, within control range; filled squares, outside control range. (**C**) Non-metric multidimensional scaling plot (Bray–Curtis, 2D stress: 0.09) based on a nontargeted analysis (https://xcmsonline. scripps.edu; *Tautenhahn et al., 2012*) of 69 chromatograms (*Figure 1—source data 1*). Color code of plant samples as in (**B**).

The online version of this article includes the following source data for figure 1:

**Source data 1.** Related to *Figure 1C*.

(two-way ANOSIM, mating status: R = −0.06, p=0.756, plant species: *R* = 0.97, p<0.0001; Bray–Curtis similarity index).

## How is plant headspace represented in the moth's antennal lobe?

In in vivo calcium imaging experiments, we successively stimulated the antennae of female *M. sexta* with puffs of the plant bouquets collected in Arizona and recorded the odor-evoked neural activity among the olfactory glomeruli of their antennal lobe. Olfactory glomeruli are functional subunits occurring in species-specific numbers. Female *M. sexta* possess 70 glomeruli arranged in a monolayer around a central neuropil (*Grosse-Wilde et al., 2011*). Activity patterns of glomeruli in the dorsal-frontal part of the antennal lobe can be monitored using in vivo calcium imaging (*Hansson et al., 2003*; *Sachse et al., 1999*). To enable comparison of headspace-evoked activation patterns among different animals, we identified 23 glomeruli in each moth using diagnostic, monomolecular odorants (*Figure 3A and B*, *Bisch-Knaden et al., 2018*). We found that plant headspace activated these 23 identified glomeruli (*Figure 3C*). However, two glomeruli (22 and 23) responded exceptionally weak. They are tuned to acids and amines (*Bisch-Knaden et al., 2018*), chemical classes that were functionally absent in the tested plant bouquets (*Figure 2C*). Next, we tested which responses were true headspace-evoked responses, that is, which responses were different from the response towards stimulations with the eluent dichloromethane (*Figure 3D*), and normalized the fluorescent signals of headspace-evoked responses for each glomerulus and animal (*Figure 3E*).

Consistent with the results from GC-EAD experiments, *Datura* and *Agave* bouquets were again unique regarding not only the number of activated glomeruli (*Figure 3D*) but also the strength of response (*Figure 3E*). *Datura* flower scent evoked the maximal response recorded in all but two glomeruli in virgin females (glomeruli 12 and 21), and in all but one glomerulus in mated females (glomerulus 12). *Agave* flower scent was the best activator for these remaining glomeruli. Apart from the weak activation levels of glomeruli 22 and 23, this exceptional representation of *M. sexta*'s two primary nectar sources was independent of the females' mating status (*Table 2*), at least among the 23 glomeruli imaged in this study.

Volatiles emitted by *M. sexta*'s larval host plants each activated only a single glomerulus out of the 23 imaged in the antennal lobe of females (*Figure 3D*). Glomerulus 4 responded to the bouquet of *Datura* foliage irrespective of the female's mating status; *Proboscidea* headspace, however, activated a different single glomerulus in virgin (glomerulus 15) than in mated females (glomerulus 12) (*Figure 3E*, *Table 2*). Furthermore, we observed a notable effect of the mating status on the representation of plants that are oviposition sites of sympatric hawkmoths (*Chilopsis*, *Helianthus*, *Vitis*). These plants evoked a major response in the antennal lobe of virgin *M. sexta* females (8–13 activated glomeruli) and an even stronger response in mated females (11–16 activated glomeruli) (*Figure 3D and E*). In contrast, females became almost anosmic towards the headspace of background plants following mating as these plants activated on average 3.1 glomeruli (range: 0–9) in virgin, but only 0.6 glomeruli (range: 0–2) in mated females (*Figure 3D*). The few glomeruli still responding towards background bouquets were different from the two host plant-activated glomeruli (*Figure 3E*, *Table 2*).

A multivariate analysis confirmed that mating status as well as plant species had a significant effect on overall activation patterns across glomeruli in the antennal lobe (two-way ANOSIM, mating status: $R = 0.71$, p=0.0001, plant species: $R = 0.83$, p=0.0001; Bray–Curtis similarity index).

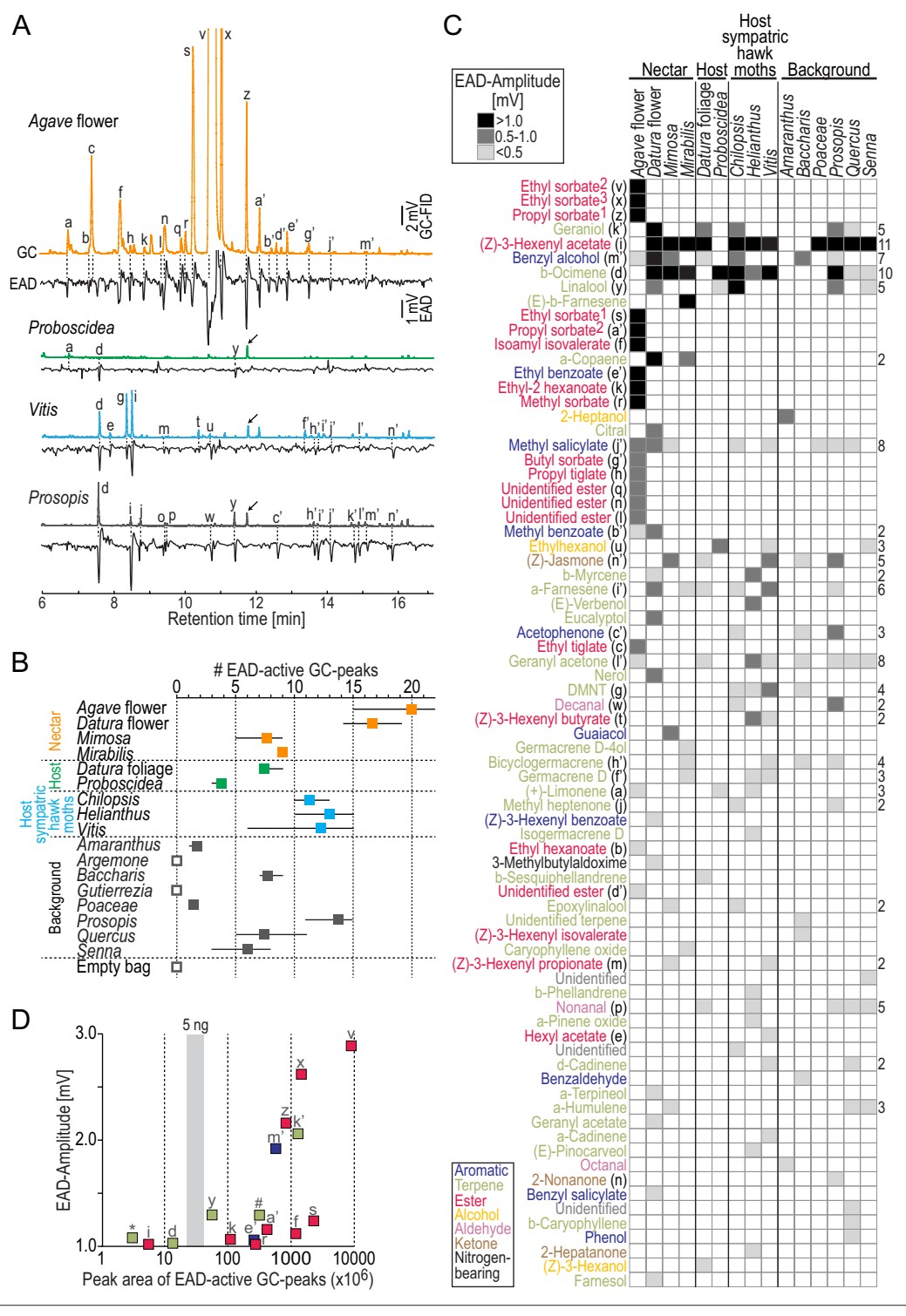

**Figure 2.** Antennal responses of *M. sexta* females to nocturnal headspaces of plants. (**A**) Examples of gas chromatography-coupled electro-antennographic detection (GC-EAD) recordings after stimulation with four plant headspaces representing nectar sources (*Agave* flower), host plants (*Proboscidea*), host plants of sympatric hawkmoths (*Vitis*), and background plants (*Prosopis*). Upper traces, gas chromatograph-coupled flame ionization detection (GC-FID); lower traces, electro-antennographic detection (EAD) of female *M. sexta*. Letters indicate EAD-active GC-peaks (labeled in **C**) that evoked a response in at least three animals. Arrows, internal standard: 5 ng 1-bromodecane; in *Agave* flower, the internal standard co-eluted with GC-peak 'z,' and GC-peaks 'v' and

*Figure 2 continued on next page*

*Figure 2 continued*

'x' are cropped. (**B**) Number of EAD-active GC-peaks per plant species. We stimulated the antennae (4–7 moths/headspace) with the same representative sample per headspace type. Filled squares, average values; whiskers, range; open squares, no active GC-peaks detected in three moths. Each moth was tested only once. (**C**) Antennal responses towards GC-peaks (rows) present in headspace (columns). Each cell in the heat map represents the median EAD amplitude of on average five moths (range: 4–7) per headspace. Rows are sorted by EAD amplitude (*Figure 2—source data 1*); magnitude of response is coded by shades of gray (see inset at top); empty cells, no response/GC fraction not present. Color code of compounds according to chemical class (see inset at bottom). Numbers next to ethyl sorbate and propyl sorbate label different enantiomers present in *Agave* flower and depict their order by retention time; DMNT, (E)–4,8-dimethyl-1,4,7-nonatriene. Numbers to the right of the heat map depict how often a given compound was present; rows without numbers indicate compounds found only in one headspace. (**D**) Effectiveness of the strongest antennal stimulants. x-axis, concentration of compounds derived from their peak area (logarithmic scale); y-axis, median EAD amplitudes ≥ 1 mV; gray vertical bar, range of peak areas of the internal standard 1-bromodecane (5 ng). For compounds present in more than one plant species, the lowest concentration eliciting a median EAD amplitude ≥ 1 mV was chosen; letters indicate compounds as in (**C**); *α-copaene; #(E)-β-farnesene. Peak area of ethyl sorbate[2] ('v') shows lower limit of concentration as the GC seemed overloaded with this odor.

The online version of this article includes the following source data for figure 2:

**Source data 1.** Related to *Figure 2C*.

## Discussion

For a female moth, two plant-based resources are of overriding importance: flowers providing nectar for sustenance of the animal itself and plants providing suitable oviposition sites and thereby food for the offspring. Here, we studied how the olfactory system of the female hawkmoth, *M. sexta*, has evolved to allow unambiguous identification of these resources based on their emissions of volatile molecules.

As could be expected, plants that predominately (*Datura*, *Mirabilis*) or at least partly (*Agave*) rely on nocturnal pollination by hawkmoths (*Alarcón et al., 2008*; *Emiliano Trejo-Salazar et al., 2015*) sent a clear and distinctive chemical signal in the night. Our results regarding the number and identity of compounds emitted by *Agave* and *Datura* flowers and the dissimilarity between both floral bouquets confirm earlier studies (*Raguso, 2004*; *Raguso et al., 2003a*; *Riffell et al., 2008*). In addition, the sunflower *Helianthus* emitted a strong and distinct scent, illustrating that its around-the-clock open flowers depend not only on diurnal but also on nocturnal pollinators. Although *M. sexta* is not described as a pollinator of *Helianthus* (*Torretta et al., 2009*), unspecified pollen from the sunflower family was found on the proboscis of *M. sexta* and other hawkmoths, indicating that these moths occasionally feed also on sunflowers (*Alarcón et al., 2008*). Low nighttime emissions observed in the remaining samples might reflect the plants' independence of nocturnal pollination (in the case of flowering plants) or an avoidance strategy against herbivory (for collections from nonflowering branches).

When we tested the antenna of female *M. sexta* with plant headspaces using GC-EAD, we found that the moths in most of the cases detect at least some compounds, even in wind-pollinated background vegetation like grass or careless weed, plants that, based on the chemical analysis, had a weak smell consisting of a small number of components. Female moths, both virgin and mated, therefore seem to be equipped with the sensory capability to distinguish not only strong and complex scents emitted by nectar sources but also bouquets of host plants and surrounding background vegetation. However, the highest number of active compounds was present in the floral headspace of *Agave* and *Datura*. In particular, the bouquet of *Agave* contained 9 of the 16 strongest antennal stimulants, 8 of them being aliphatic esters. These esters are signature compounds of *Agave* flowers (*Raguso, 2004*) as they are rarely found in other floral headspace investigated in almost 1000 plant species from 90 families (*Knudsen et al., 2006*). In particular, this chemical class is lacking in typical hawkmoth-pollinated flowers (*Raguso et al., 2003a*; *Raguso et al., 2003b*). Two enantiomers of the *Agave*-characteristic ester ethyl sorbate elicited the strongest antennal response of all active GC-peaks (median EAG amplitudes: 2.6 mV and 2.9 mV, stimulus concentrations: ~0.5 µg and >>0.5 µg, *Figure 2D*). These responses are higher than the response of a male antenna when stimulated with bombykal, the main component of *M. sexta*'s sex pheromone (1.9 mV, stimulus concentration: ~100 µg; *Fandino et al., 2019*). Even if we consider that bombykal has a lower vapor pressure than ethyl sorbate and that less

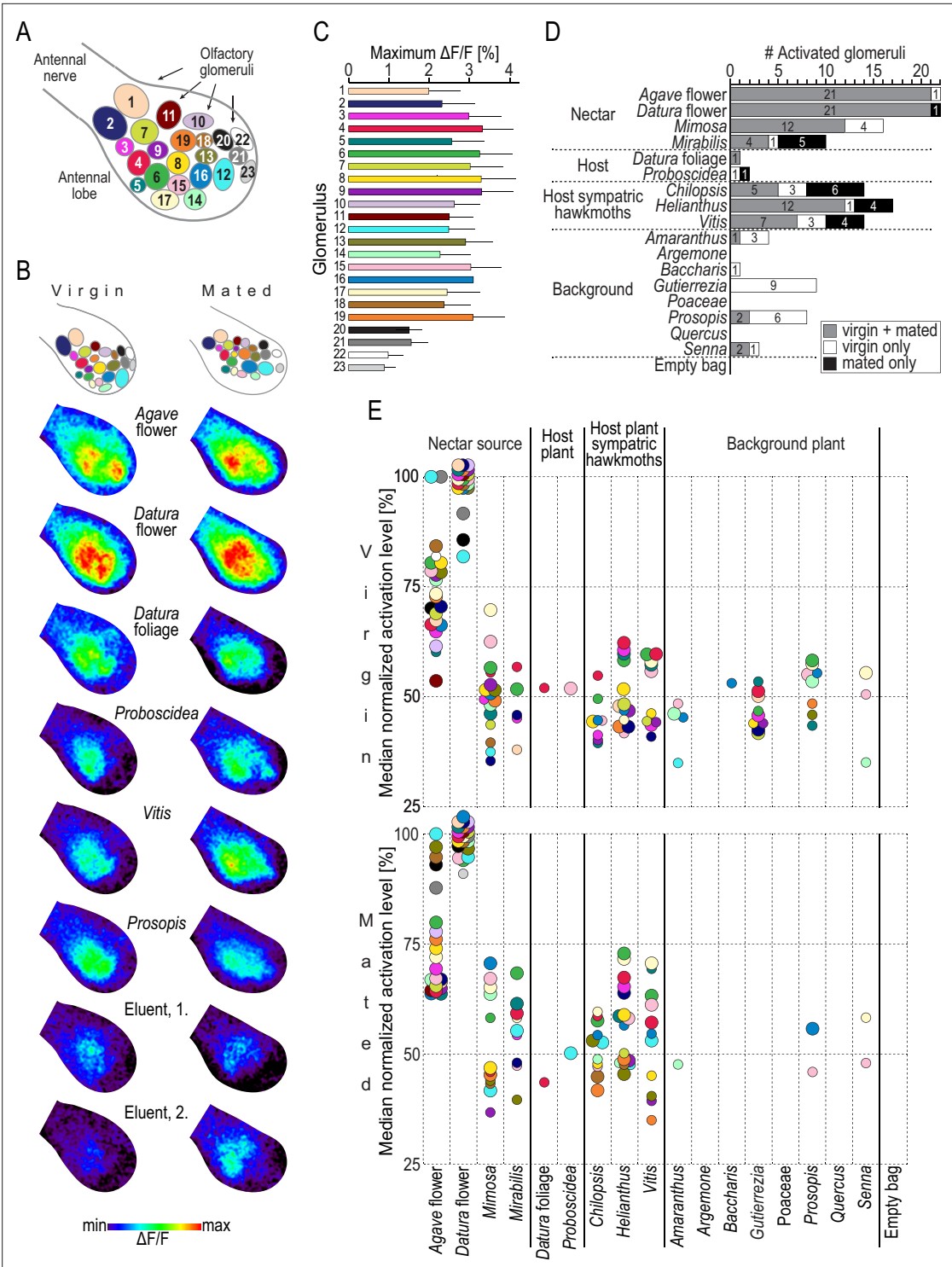

**Figure 3.** Headspace-evoked activity patterns in the antennal lobe of female *M. sexta*. (**A**) Schematic of 23 olfactory glomeruli at the dorsal surface of the right antennal lobe. Entrance of the antennal nerve is in the upper-left corner. Numbers, glomeruli identification as in *Bisch-Knaden et al., 2018*. (**B**) Examples of in vivo calcium imaging recordings after stimulation with plant headspaces representing nectar sources (*Agave* flower, *Datura* flower), host plants (*Datura* foliage, *Proboscidea*), host plants of sympatric hawkmoths (*Vitis*), background plants (*Prosopis*), and the eluent dichloromethane (first and second stimulations at the beginning and end of the experiment). False-color-coded imaging results of the right antennal lobe in a virgin (left column) and a mated female (right column) normalized to their highest response (see color bar). Top row, schematic of individual antennal lobes, colors as in (**A**). (**C**) Maximum increase of fluorescence in 23 identified glomeruli. Graph depicts for each glomerulus (color code as in **A**) the average maximum responses (bars) and 1 standard deviation (whiskers) of 10 virgin and 10 mated females after stimulation with plant headspaces. In 69% of

*Figure 3 continued on next page*

*Figure 3 continued*

460 cases (20 maximum values in 23 glomeruli), *Datura* flower was the headspace eliciting the maximum response, and in 17% it was *Agave* flower. (**D**) Number of activated glomeruli in the antennal lobe depending on female mating status. A glomerulus was scored as activated if its headspace-evoked response was different from the averaged response to the two stimulations with the eluent dichloromethane (p<0.01, Friedman test with Dunn's multiple-comparisons test). For the identity of glomeruli activated by each plant headspace, see *Table 2*. (**E**) Activity levels evoked by plant headspace in individual glomeruli in the antennal lobe. Colored dots represent median normalized responses of activated glomeruli in 10 virgin (top) and 10 mated (bottom) females; color code of glomeruli as in (**A**). Only values of activated glomeruli are shown (small circles, p<0.01, large circles, p<0.001, Friedman test with Dunn's multiple comparisons test, *Figure 3—source data 1*).

The online version of this article includes the following source data for figure 3:

**Source code 1.** Custom-written software for processing calcium imaging data in IDL (L3Harris Geospatial).

**Source data 1.** Calcium imaging results from 10 virgin and 10 mated females (*Figure 3E*).

odor might reach the antenna in EAG than in GC-EAD experiments, this comparison indicates that the female antenna is at least as sensitive to promising floral, that is, nectar-indicating volatiles, as the male antenna by the female sex-pheromone.

EAD activity might correlate with behavior (*Liu et al., 2020*; *Zhu et al., 1993*), but a strong antennal response towards an odor does not always imply a strong behavioral response to this odor molecule

**Table 2.** Headspace-activated glomeruli independent and dependent of mating status.

| Glomerulus | Response independent of mating status | Response only before mating | Response only after mating |
|---|---|---|---|
| 1* | Agave, Datura | Mirabilis, *Helianthus*, Gutierrezia | |
| 2 | Agave, Datura, Mirabilis, *Helianthus* | Mimosa, *Vitis*, Gutierrezia | |
| 3 | Agave, Datura, Mirabilis, *Helianthus* | Mimosa, *Chilopsis*, *Vitis*, Gutierrezia | |
| 4* | Agave, Datura, Mimosa, Mirabilis, <u>Datura foliage</u>, *Chilopsis*, *Helianthus*, *Vitis* | Gutierrezia | |
| 5 | Agave, Datura, Helianthus, *Vitis* | Mimosa, *Chilopsis*, Gutierrezia, Prosopis | Mirabilis |
| 6* | Agave, Datura, Mimosa, Mirabilis, *Chilopsis*, *Helianthus*, *Vitis* | Gutierrezia, Prosopis | |
| 7 | Agave, Datura, *Helianthus* | Mimosa, *Vitis*, Gutierrezia | |
| 8 | Agave, Datura, Mimosa, *Chilopsis*, *Helianthus*, *Vitis* | Gutierrezia | |
| 9 | Agave, Datura, Mimosa, *Helianthus*, *Vitis* | *Chilopsis*, Gutierrezia | |
| 10 | Agave, Datura | | |
| 11 | Agave, Datura | | |
| 12* | Agave, Datura, Mimosa | Amaranthus | Mirabilis, <u>Proboscidea</u>, *Chilopsis*, *Helianthus*, *Vitis* |
| 13* | Agave, Datura, Mimosa | Prosopis | Mirabilis, *Chilopsis*, *Helianthus*, *Vitis* |
| 14 | Agave, Datura, Mimosa, Amaranthus | Prosopis, Senna | *Chilopsis*, *Helianthus* |
| 15 | Agave, Datura, Mimosa, *Chilopsis*, *Helianthus*, *Vitis*, Prosopis, Senna | <u>Proboscidea</u>, Amaranthus | Mirabilis |
| 16 | Agave, Datura, Mimosa, *Chilopsis*, *Helianthus*, Prosopis | Amaranthus, Baccharis | *Vitis* |
| 17* | Agave, Datura, Mimosa, *Helianthus*, *Vitis*, Senna | Prosopis | Mirabilis, *Chilopsis* |
| 18* | Agave, Datura, Mimosa | | *Chilopsis*, *Helianthus* |
| 19* | Agave, Datura, Mimosa, *Helianthus* | Prosopis | *Chilopsis*, *Vitis* |
| 20* | Agave, Datura | | |
| 21* | Agave, Datura | | |
| 22 | | Agave | |
| 23 | | | Datura |

Font format depicts type of plant headspace: nectar source of *M. sexta*, host plant of *M. sexta*, *host plant of sympatric hawk moths*, <u>background plant</u>.
*Glomerulus whose activation level is positively correlated with odor-guided behavior of virgin females in wind tunnel experiments (***Bisch-Knaden et al., 2018***).

(*Honda et al., 1998*; *Suckling et al., 1996*). In *M. sexta*, a comparison of physiological and behavioral data is possible as almost half of the active and identified GC-peaks in this study were previously tested in a wind tunnel assay (*Bisch-Knaden et al., 2018*). EAD responses evoked by these 31 shared odors belonging to seven chemical classes are indeed positively correlated with the duration a female moth shows feeding behavior when encountering the same odors (EAD amplitude versus duration of proboscis contacts with a scented filter paper, Pearson correlation coefficient $r = 0.41$, p=0.023). In contrast, no correlation was found between EAD activity and the duration of abdomen curling behavior, that is, behavior related to oviposition (r = −0.03, p=0.9). Hence, no conclusions from GC-EAD results can be drawn regarding an odor's relevance in connection with an oviposition site, whereas odors that evoked a strong response at the antenna of female *M. sexta* often are attractive in the context of feeding.

In previous GC-coupled single-sensillum recordings (GC-SSR), 60% of randomly chosen olfactory sensilla on the antenna of female *M. sexta* reacted to the aliphatic ester (Z)-3-hexenyl acetate when stimulated with the scent of herbivore-damaged *Datura* foliage (*Spaethe et al., 2013b*). If the olfactory sensory neurons housed in these sensilla would not only detect (Z)-3-hexenyl acetate but aliphatic esters in general, this could explain the prominent response towards typical *Agave* esters in our GC-EAD experiments. In addition, the antenna might harbor narrowly tuned olfactory sensory neurons strongly responding only to *Agave* esters. Hawkmoth-pollinated flowers like *Datura*, *Nicotiana*, and *Petunia* emit oxygenated aromatics that are especially attractive to foraging hawkmoths and elicit a strong response from the antenna of *M. sexta*, and in its antennal lobe, respectively (*Bisch-Knaden et al., 2018*; *Hoballah et al., 2005*; *Kessler et al., 2008*; *Riffell et al., 2013*). This study shows that *M. sexta* females in addition exhibit a robust physiological response towards the bouquet collected from *Agave* flowers, reflecting the significant role this copious nectar source — releasing a very different smell than typical hawkmoth flowers — plays in *M. sexta*'s foraging behavior. Other hawkmoth species feeding on nectar from *Agave* flowers might share similar olfactory detection and processing abilities (*Alarcón et al., 2008*; *Emiliano Trejo-Salazar et al., 2015*).

Three volatiles, α-copaene, (Z)-3-hexenyl acetate, and β-ocimene, stood out as particularly strong activators of the female *M. sexta* antenna, although they were present in very low concentrations. The high sensitivity towards these odors might indicate that they act as long-distance cues guiding the moth to places with vegetation (*Webster and Cardé, 2017*). Furthermore, these odors have meanings that are more specific: α-copaene is involved in the oviposition decision process of *M. sexta* (*Zhang et al., 2022*) and in addition might indicate rewarding nectar sources as it was functionally present, that is, EAD-active, only in the headspace of *Datura* flower and *Mirabilis*. (Z)-3-hexenyl acetate and β-ocimene, on the other hand, are typical herbivore-induced volatiles and are released by herbivore-damaged *Datura* leaves (*Allmann et al., 2013*; *Hare and Sun, 2011*; *Zhang et al., 2022*). Interestingly, some projection neurons in the female antennal lobe targeting an identified glomerulus were reported to specifically respond to low concentrations of (Z)-3-hexenyl acetate (*Reisenman et al., 2005*). This odor might thus inform a *M. sexta* female searching for oviposition sites about the presence of potential larval competitors and predators already at a distance.

An earlier GC-EAD study with female *M. sexta* reported that *Datura* and *Proboscidea* foliage each emit 10 identified EAD-active compounds and share 8 of them (*Fraser et al., 2003*). Our work, in contrast, shows that both host plants emit only nine and four active compounds, respectively, and have no active compounds in common. In the case of *Proboscidea* headspace, none of its EAD-active compounds found in our experiments was identified in the former study, and vice versa. Many methodological factors could have contributed to this discrepancy in the results. In detail, Fraser et al. collected headspace from single, potted, undamaged plants with buds, flowers, and seeds removed; and a cultivar of *Proboscidea* was used, not the wild type. In this study, we collected headspace of local, mostly herbivore-damaged plants growing in a natural plant community and did not remove any parts of the plant. In addition, our odor collection lasted 12 hr during the natural dark phase versus 24 hr of artificially induced scotophase in *Fraser et al., 2003*. Conditions like growth in mixed plant populations or in monocultures (*Kigathi et al., 2019*), light deprivation (*He et al., 2021*), herbivore attack, and other stress factors (*Holopainen and Gershenzon, 2010*) influence both composition and quantity of plant-emitted volatiles. Therefore, the observed differences between the studies could be expected and emphasize the significance of odor collections in the field.

Bath application of a fluorescent calcium sensor allows monitoring of odor-induced neural activity in the brain. Each neuron type in the treated brain region might take up the marker molecules. However, as each glomerulus in the antennal lobe receives input from 4000 to 5000 olfactory sensory neurons (*Oland and Tolbert, 1988*), and is targeted by only 4–5 projection, that is, output neurons (*Homberg et al., 1988*), odor-evoked activation patterns in calcium imaging experiments can be assumed to reflect mainly the activity of input neurons. Additionally, about 360 local interneurons per antennal lobe (*Homberg et al., 1988*) with inhibitory and/or excitatory functions (*Reisenman et al., 2011*) might synapse back onto the sensory neurons, thus modulating their activity and accordingly the observed calcium signal. Although most of these interneurons arborize in many, if not all, glomeruli, some interneurons have a more restricted innervation pattern and connect only a few glomeruli (*Christensen et al., 1993*). This type of interneuron seems predisposed to play a role in the coding of complex odor blends released by plants. Interestingly, patchy interneurons are present mainly in female *M. sexta* (*Matsumoto and Hildebrand, 1981*). In the vinegar fly *Drosophila melanogaster*, patchy interneurons are responsible for nonlinear processing of binary odor mixtures (*Mohamed et al., 2019*). For some glomeruli in *D. melanogaster*, this modulation occurred already at the presynaptic level, that is, at the level we monitored in our calcium imaging experiments. To estimate if nonlinear interactions might occur in the antennal lobe of *M. sexta*, we compared headspace-evoked activation patterns with activation patterns evoked by EAD-active, single compounds that were present in the respective headspace. From the bouquet of *Datura* foliage, for example, (Z)-3-hexenyl acetate elicited the strongest antennal response (*Figure 2C*) and activates mainly four glomeruli (glomeruli 6, 13, 16, and 12) when tested on its own (*Bisch-Knaden et al., 2018*). After stimulation with the complex headspace, however, none of these glomeruli was responding (*Figure 3E*, *Table 2*). The second best antennal activator in the headspace of *Datura* foliage, geraniol, activates mainly three glomeruli (glomeruli 6, 4, and 5, *Bisch-Knaden et al., 2018*). Of these, glomerulus 4 was the only one responding towards stimulation with the headspace (*Figure 3E*, *Table 2*). We thus conclude that there are indications of local inhibition as we otherwise would have expected to observe more activated glomeruli after stimulation with the complex headspace. A similar inhibition of glomeruli in mixtures of odors was reported in a calcium imaging study in honey bees, where the inhibitory effect was stronger in ternary than in binary mixtures (*Joerges et al., 1997*). As the plant bouquets tested in our study contained up to 20 EAD-active components, and as local interneurons in *M. sexta*, like in most insects, are mainly inhibitory (*Christensen et al., 1993*), the observed inhibitory mixture interactions after stimulation with complex blends seem plausible.

However, we also revealed coding characteristics that were similar at the periphery and in the brain, especially after stimulation with feeding-related odor bouquets. Representation of the essential nectar sources *Datura* and *Agave* flowers in the antennal lobe was outstanding as in the vast majority of all glomerulus-headspace combinations these two floral scents elicited the highest response. In former studies using a different approach, only a small fraction of the compounds present in the two floral bouquets (*Agave*: 10%; *Datura*: 15%) activated neurons in the antennal lobe of male *M. sexta* (*Riffell et al., 2009a*; *Riffell et al., 2009b*). However, the recording technique used by Riffell and colleagues targets about 10 glomeruli and part of the adjacent neuropil in the lateral part of the male antennal lobe, while we recorded from 23 identified glomeruli in the dorsal part of the female lobe. In addition, we used a puff of the floral headspace as stimulus, that is, the antennal lobe was activated by the full floral blend, whereas in previous studies a GC-coupled stimulus was applied, that is, the antennal lobe was activated by temporally separated single compounds present in the floral blend. The differing results might thus be due to these different approaches and potential sex-specific coding differences. However, compounds that were identified to activate neurons in the male antennal lobe, like ethyl sorbate (*Agave*) and benzyl alcohol (*Datura*), were also EAD-active in this study (*Figure 2C*) and evoked responses in some glomeruli of the female antennal lobe in a previous imaging study (*Bisch-Knaden et al., 2018*). Interestingly, lab experiments in a wind tunnel suggest that even such a strong scent as the one from *Datura* flowers can become less attractive to male *M. sexta* when presented in the olfactory background of a selected nonhost plant (*Riffell et al., 2014*).

Olfactory coding can change depending on the mating status of an insect (*Gadenne et al., 2016*). However, the neural response of female *M. sexta* towards volatiles from its main nectar sources was not altered following mating as it was reported for the noctuid moth *Spodoptera littoralis* (*Saveer et al., 2012*). Different life history traits of noctuid and sphingid moths might explain this different

result: noctuid moths are generalists and lay their eggs in clusters on a wide range of acceptable host plants. Sphingid moths like *M. sexta*, on the other hand, are usually specialized on a few host plant families and females lay only a few single eggs on a given plant. Thus, hawkmoths need to refill their energy reservoir between oviposition bouts at host plants that are rare in the habitat and therefore require long flights between them (*Alarcón et al., 2008*; *Raguso et al., 2003a*). Moreover, female hawkmoths benefit from nectar feeding following mating as they live longer and produce more mature eggs compared to starved females (*Sasaki and Riddiford, 1984*; *von Arx et al., 2013*). The energy demand of hawkmoths is in addition especially high as both feeding and oviposition usually occur while the moth is hovering in front of the plant (*Stöckl and Kelber, 2019*). Taken together, the prominent and mating status-independent representation of floral bouquets at the antenna and in the antennal lobe of female *M. sexta* is in accordance with the moths' ecology.

While the coding of flower volatiles in nectar-feeding moths is probably independent of sex, odors indicating oviposition sites should be of special importance for female moths after mating. Two enlarged, female-specific glomeruli that are located at the entrance of the antennal nerve into the female antennal lobe — at the same position as the sex pheromone-processing macroglomerular complex in males (*Matsumoto and Hildebrand, 1981*; *Rössler et al., 1998*) — seem predisposed to be involved in oviposition choice. This hypothesis is supported by the fact that output neurons targeting both glomeruli respond to headspace of tomato leaves, another host plant for *M. sexta* (*King et al., 2000*; *Reisenman et al., 2009*). On the other hand, the two host plant bouquets tested in our imaging experiments did not activate these glomeruli (glomeruli 1 and 2, *Table 2*), confirming results of a study using vegetative headspace from the hosts *Datura*, *Nicotiana*, and tomato. These scents failed to evoke a response in sensilla targeting mainly the two female-specific glomeruli (*Shields and Hildebrand, 2001*). Therefore, the question if these glomeruli might be involved in identifying an oviposition site is still open.

In contrast to the wide and strong activation of antennal lobe glomeruli by flower odors, *M. sexta*'s host plant bouquets each activated only a single glomerulus of the 23 glomeruli under investigation. While the responding glomerulus towards *Datura* foliage was independent of the female's mating status, *Proboscidea* activated a different glomerulus in virgin than in mated females. This result is in line with the fact that the ecological meaning of *Datura* foliage does not change after mating as its smell indicates both a suitable host plant and a profitable nectar source (*Kárpáti et al., 2013*). *Proboscidea*, on the other hand, does not provide nectar for hawkmoths and is therefore interesting for the female moth only after mating. Many EAD-active compounds were tested in a previous calcium imaging study using monomolecular odorants as stimuli (*Bisch-Knaden et al., 2018*), allowing a comparison between these data and our imaging results obtained with natural mixtures. Some compounds present in the headspace of *Proboscidea* and *Datura* foliage, for example, when tested alone activated most strongly glomerulus 6, a glomerulus that was not activated after stimulation with the complete headspaces, again indicating nonlinear processing and robust presynaptic inhibitory interactions between glomeruli (*Joerges et al., 1997*; *Mohamed et al., 2019*). The two host plant-activated glomeruli in mated females were as well responding to nectar sources and hosts of sympatric hawkmoths. However, host plants exclusively activated one of these glomeruli, whereas the other sources activated additional glomeruli. Even if nonhost plants as well as host plants would activate more glomeruli in areas of the antennal lobe that were inaccessible in our imaging study, the resulting neural representation of nonhost plants in the antennal lobe of mated females would remain different from the pattern evoked by host plants.

Interestingly, virgin and mated females differed markedly in their response to the odor of background plants: of the 23 identified glomeruli, these plants activated only a small number in virgin females (range: 0–9) and even less glomeruli (range: 0–2) in mated females. The few background-activated glomeruli did not include the two host plant-activated glomeruli. The moths' reduced responses to background plants but not to host plants after mating, together with the finding that the host plant *Proboscidea* activates a different glomerulus in virgin and mated females, indicate that the olfactory system of *M. sexta* females becomes tuned towards host plants following mating. Mechanisms mediating these post-mating changes in moth olfactory processing seem to be independent of neurotransmitters like octopamine and serotonin (*Barrozo et al., 2010*) but might include neuropeptides as in the vinegar fly, *D. melanogaster* (*Hussain et al., 2016*), and regulation of chemosensory-related genes like in *Drosophila suzukii* (*Crava et al., 2019*). Our data suggest that mated females

could potentially be able to identify suitable oviposition sites by the relative activity of one or two glomeruli compared with the activity of other glomeruli. A similar sparse coding strategy was recently described for the discrimination of differentially attractive body odors by mosquitoes (*Zhao et al., 2020*). In this case, the relative activity of a single, human-odor-activated glomerulus versus a broadly tuned glomerulus has been proposed to enable the mosquito to identify its preferred human host.

*M. sexta* females intersperse feeding and oviposition bouts when visiting a flowering *Datura* (*Raguso et al., 2003a*), and lay more eggs on flowering than on nonflowering plants (*Reisenman et al., 2010*). However, *Datura* foliage alone attracts egg-laying females in the field (personal observation) and the lab (*Nataraj et al., 2021*; *Spaethe et al., 2013b*). We therefore tested leaves of *Datura* separately and compared their headspace with that of *Proboscidea*, the only other host plant in our study area. As the single activated glomerulus was different after stimulation with the two host plant bouquets, which also did not share any EAD-active compounds, *M. sexta* females should be able to distinguish the two plants based on olfaction alone. Field observations and experiments show that females lay more eggs on *Proboscidea* than on solanaceous hosts, although the plants grow next to each other and have a similar leaf surface (*Diamond and Kingsolver, 2010*; *Mechaber and Hildebrand, 2000*). These findings indicate that *M. sexta* can indeed discriminate between host plants belonging to different plant families, although visual and tactile cues might play a role in combination with olfactory cues. The observed low overall activity across the antennal lobe evoked by host plant odors corresponds to the preference of *M. sexta* for plants with a faint smell when searching for oviposition sites. Inbred horse nettle (*Solanum carolinense*) exhibits much lower nocturnal volatile emissions than outbred horse nettle, a solanaceous host plant of *M. sexta* in the southeastern US. Correspondingly, female moths spend more time hovering near inbred plants and lay more eggs there than on outbred plants. This preference is governed by olfactory cues alone as it persists in the absence of visual and contact cues (*Kariyat et al., 2013*). Furthermore, when given the choice between headspaces of two solanaceous host plant species with different total volatile concentration, *M. sexta* clearly prefers the weaker smelling plant. Diluting the headspace of the more intensely smelling plant leads to a reduction in this preference (*Spaethe et al., 2013a*). These findings show again that female moths consistently favor host plants with low volatile emissions, probably because high emission of specific volatiles are signs of active plant defense mechanisms, indicating the presence of larval competitors (*De Moraes et al., 2001*), and leading to impaired larval growth (*Delphia et al., 2009*). High levels of these herbivore-induced volatiles also attract predators and parasitoids (*Kessler and Baldwin, 2001*; *Turlings et al., 1995*), and egg-laying moths therefore avoid these sites (*De Moraes et al., 2001*; *Li et al., 2018*). Conversely, when *M. sexta* has to choose between flowering tobacco plants from populations that differ in their flower volatile concentration, the moths clearly prefer to forage at flowers with a stronger smell (*Haverkamp et al., 2018*). Hence, *M. sexta* pursues different strategies when searching for oviposition or feeding sites as the moths favor weakly or strongly scented sources, respectively.

In contrast to *M. sexta*'s host plants, the bouquets of host plants of sympatric hawkmoths activated many glomeruli in virgins, and even more glomeruli in mated females. Two glomeruli contributed mostly to this effect as they were responding to all three nonhost bouquets only after mating (glomeruli 12 and 13; *Table 2*). Odor-induced activation levels of these two glomeruli were positively correlated to odor-induced behavior in wind tunnel experiments using monomolecular odorants (*Bisch-Knaden et al., 2018*). However, only virgin females were included in this study, so conclusions regarding the behavior of mated females cannot be drawn.

The strong activation of antennal lobe glomeruli by host plants of other hawkmoths living in the same habitat was in contrast to weak but specific activation of single glomeruli (among those imaged) by host plants of *M. sexta*. The conspicuous activation patterns evoked by host plants of sympatric hawkmoths might serve as a stop signal for *M. sexta* during their search for a suitable oviposition site and therefore might help gravid females to avoid inappropriate hosts at a distance. It would be interesting to compare headspace-evoked activation patterns in the antennal lobe of co-occurring hawkmoths upon stimulation with odors of their own and of other species' host plants to test if this might be a general coding policy. Examples of olfaction-based avoidance of nonhost plants were also reported for example in bark beetles (*Huber et al., 2000*). Antennae of these insects respond strongly to many volatiles released by nonhost trees. Like in the case of *M. sexta*, some compounds are present in the bouquet of both host and nonhost plants, corroborating the hypothesis that odor-guided choice

of host plants relies on blends of ubiquitous compounds in a specific ratio (*Bruce and Pickett, 2011*) and concentration (*Spaethe et al., 2013a*) rather than on the detection of host-exclusive odors.

By using ecologically relevant odors collected in the actual habitat of our model animal, *M. sexta*, we revealed olfactory coding strategies both for odors emanating from crucial resources but also for those emitted by substrates that should be avoided. We also show how the female mating status affects olfactory processing but, interestingly, in a way well adapted to the specific life history traits of the species under investigation. In a broader perspective, our study contributes to understanding innate neural representation of natural odor mixtures in the brain and coding strategies enabling animals to distinguish crucial resources from background noise.

## Materials and methods

### Headspace collection in the field

We collected the headspace of plants in a habitat of *M. sexta* at the Santa Rita Experimental Range, 40 km south of Tucson, Arizona, at the foot of the Santa Rita Mountains (31°78′ N, 110°82′ W). All plant species sampled (*Table 1*) are native to the habitat and belong to the regular desert grassland vegetation at Santa Rita Experimental Range (*Medina, 2003*). We sampled from flowering plants or flowering branches if the respective plant was blooming during the experimental nights. Otherwise, nonflowering branches were sampled (*Table 1*). *Agave* is a succulent plant with a basal rosette of sharp-edged leaves, each with a length of c. 1 m and long spines at the tip. These leaves did not fit in our collection bags (see below). We therefore only collected headspace from *Agave* flowers, which appear in umbels at the end of a long bloom stalk without leaves (about 5–6 m above the basal rosette of leaves). In the case of *Datura*, the flower is a valuable nectar source and the leaves are an oviposition substrate for *M. sexta*. We therefore tested flowers and leaves of *Datura* separately.

At sunset, we carefully enclosed plants in polyethylene terephthalate bags (Toppits, Germany). Charcoal-filtered, environmental air was pumped into the bag through a silicone tube connected to a custom-made portable pump. Air was pumped out of the bag through a second silicone tube passing a volatile collection trap (Porapak-Q 25 mg, https://www.volatilecollectiontrap.com). Shortly after sunrise, we unpacked the plants, removed the volatile collection traps, and stored them at −20°C. We collected the headspaces of plants on nine consecutive nights (August 19–27, 2018). In the first and last nights, we made a control collection with an empty bag placed on the ground close to the collection sites of plant headspaces. One volatile collection trap not used but treated in the same way as headspace-collecting traps served as a handling control. In Jena, Germany, all volatile collection traps were eluted with 4 × 100 µl dichloromethane containing 5 ng/µl bromodecane as an internal standard.

### Chemical analysis

Headspace samples were analyzed by GC-MS (7890B GC System, 5977A MSD, Agilent Technologies, https://www.agilent.com) equipped with a polar column (HP-INNOWAX, 30 m long, 0.25 mm inner diameter, 25 µm film thickness; Agilent) with helium as carrier gas. The inlet temperature was set to 240°C. The temperature of the GC oven was held at 40°C for 3 min, and then increased by 10°C per min to 260°C. This final temperature was held for 15 min. The MS transferline was held at 260°C, the MS source at 230°C, and the MS quad at 150°C. Mass spectra were taken in electron ionization mode (70 eV) in the range from *m/z* 29 to 350. GC-MS data were processed with the MDS-ChemStation Enhanced Data Analysis software (Agilent).

### Breeding of *M. sexta*

*M. sexta* larvae were reared in the laboratory on artificial diet (*Grosse-Wilde et al., 2011*). Female pupae were kept in a climate chamber (25°C, 70% relative humidity) with a reversed light cycle (8 hr dark/16 hr light), and moths were tested during their scotophase on days 2–4 after hatching (GC-EAD), or at day 3 after hatching (calcium imaging). Moths were unfed and had no experience with plant-derived volatiles. To obtain mated females, we placed them in a cage with an equal number of males 1 day before an experiment was planned. We checked the cage 3–4 hr later and removed all animals that were not mating.

### GC-EAD recordings

We used GC with flame-ionization detection (GC-FID) coupled with electro-antennographic detection (EAD) to identify compounds in headspace collections that can be sensed by *M. sexta*. One antenna of

a female moth (virgin and mated in equal numbers), age 2–4 days, was cut and connected to two glass electrodes filled with physiological saline solution (*Christensen and Hildebrand, 1987*). The reference electrode was inserted into the basal segment of the antenna, and the recording electrode was brought in contact with the tip of the antennae. The EAD signal (transferred via Ag-AgCl wires) was pre-amplified (10×) with a probe connected to a high-impedance DC amplifier (EAG-probe, Syntech, https://www.syntech.nl) and digitally converted (IDAC-4 USB, Syntech), visualized, and recorded on a PC using the software Autospike (Syntech). For each run, 2 μl of a headspace sample (a lower amount of 1 μl for *Agave* flower and *Datura* flower) was injected into a GC-FID (6890N, Agilent) equipped with a polar column (HP-INNOWAX, 30 m long, 0.32 mm inner diameter, 0.25 μm film thickness, Agilent) with helium as carrier gas. The inlet temperature was set to 250°C. The temperature of the GC oven was held at 40°C for 2 min, and then increased by 10°C per min to 260°C. This final temperature was held for 10 min. The GC was equipped with an effluent splitter (Gerstel) at the end of the analytical column, with a GC:antenna split ratio of 1:10 and helium as carrier gas. One arm was connected with the FID of the GC, and the other arm entered a heated (270°C) GC-EAD interface (Syntech) that was connected to a bent glass tube (diameter: 12 mm). The antenna-directed GC effluent was mixed with a humidified, charcoal-filtered air stream (1 l/min) to cool the effluent down and guide it to the antenna. Signals from the moth's antenna and the FID were recorded simultaneously. Sample size was 4–7 antenna per sample type; if a given sample type did not elicit a single response in three different animals, it was not tested any further (*Argemone*, *Gutierrezia*, empty bag). A GC-peak was scored as EAD-active if it induced a response at the same retention time in at least three antennae and if this GC fraction was present in at least one other headspace of the same type.

## Identification of compounds

On both GC instruments (GC-MS and GC-FID), we ran a series of 20 n-alkanes and matched retention times of EAD-active peaks and peaks obtained with the GC-MS using their Kovats retention indices. EAD-active peaks were tentatively identified by comparison of their mass spectra and Kovats retention indices with those from a reference library (National Institute of Standards and Technologies), and a database built in our laboratory with synthetic standards using the same GC-MS instrument. Compounds yielding a match of mass spectra above 90% were rated as tentatively identified. The fragmentation pattern of some EAD-active peaks could not be clearly matched to any library compound and was labeled as unidentified (see *Figure 2C*).

## Preparation for calcium imaging experiments

Female moths were tested; they were either virgin or mated on day 1 after emergence. On day 2, moths were positioned in a 15 ml plastic tube with the tip cut open. The head was protruding at the narrow end and was fixed in this position with dental wax. Labial palps and proboscis were also fixed with wax to reduce movement artifacts during the experiments. A window was cut in the head capsule between the compound eyes, and the tissue covering the brain was removed until the antennal lobes were visible. We added 50 μl Pluronic F-127 (Invitrogen) to 50 μg of the membrane-permeant form of a fluorescent calcium indicator (Calcium Green-1 AM, Invitrogen) and sonicated the solution for 10 min. Then, we added 800 μl physiological saline solution (*Christensen and Hildebrand, 1987*) and sonicated again for 10 min. Then, 20 μl of this dye solution was applied to the exposed brain, and the preparation was incubated in a humid chamber for 45 min at room temperature. Then, we rinsed the brain several times with physiological saline solution to remove excess dye and stored the moths at 4°C overnight to calm them down and reduce their movements. Imaging experiments were performed the following day (day 3 after emergence).

## Calcium imaging

The imaging setup consisted of a CCD camera (Olympus U-CMAD3) mounted to an upright microscope (Olympus BX51WI) equipped with a water immersion objective (Olympus, 10×/0.30). Calcium green-1 AM was excited at 475 nm (500 nm shortpass optical filter; xenon arc lamp, Polychrome V, Till Photonics), and fluorescence was detected at 490/515 nm (dichroic longpass/longpass). The setup was controlled by the software Tillvision version 4.6 (Till Photonics). Fourfold symmetrical binning resulted in image sizes of 344 × 260 pixels, with one pixel corresponding to an area of 4 μm × 4 μm.

## Odor stimulation

To create a functional map of glomeruli in the antennal lobe, we first tested 19 diagnostic odors (*Bisch-Knaden et al., 2018*) in each animal. Then, we tested 17 headspaces of plants (*Table 1*) and 1 collection with an empty bag. The same samples as in GC-EAD experiments were used. Then, 10 µl of a diagnostic odor or an eluted headspace were applied onto a circular piece of filter paper (diameter: 12 mm, Whatman) that was inserted in a glass pipette; 10 µl of the solvent mineral oil (diagnostic odors) or the eluent dichloromethane (headspace) served as control stimuli. A pipet tip sealed with dental wax closed the pipettes until the start of the experiment. As dichloromethane alone evoked a response in the antennal lobe (*Figure 3B*), pipettes with headspace samples and two pipettes with dichloromethane were left open for 3–5 min before sealing them to allow evaporation of the eluent. Filter papers were renewed every experimental day (diagnostic odors) or the pipettes were stored at –20°C and used on up to three experimental days (headspace). The immobilized moth was placed upright under the microscope. A glass tube (diameter: 5 mm) was directed perpendicular to one antenna and delivered a constant stream of charcoal-filtered, moistened air (0.5 l/min). Two glass pipettes were inserted through small holes in the tube. One pipette (inserted 5.5 cm from end of tube) was empty and added clean air to the continuous airstream (0.5 l/min). This airstream could automatically be switched (Syntech Stimulus Controller CS-55) to the second pipette (inserted 3.5 cm from the end of tube) that contained an odor-laden filter paper. By this procedure, the airstream reaching the antenna was not altered during odor stimulation, thus reducing mechanical disturbances. One odor stimulation experiment lasted 10 s and was recorded with a sampling rate of 4 Hz corresponding to 40 frames. The time course of an odorant stimulation experiment was as follows: 2 s clean airstream (frames 1–8), 2 s odorous airstream (frames 9–16), and 6 s clean airstream (frames 17–40). Odors were presented with at least 1 min interstimulus interval to avoid adaptation. The sequence of headspace stimulations changed from animal to animal, and a control stimulus with dichloromethane was presented at the beginning and end of this sequence.

## Processing of calcium imaging data

Stimulation experiments resulted in a series of 40 consecutive frames that were analyzed with custom-written software (IDL, L3Harris Geospatial; *Figure 3—source code 1*). Several processing steps were applied to enhance the signal-to-noise ratio: (1) background correction: background activity was defined as the average fluorescence (F) of frames 3–7 (i.e., before stimulus onset) and was subtracted from the fluorescence of each frame. This background-corrected value (deltaF) was divided by the background fluorescence to get the relative changes of fluorescence over background fluorescence for each frame (deltaF/F). (2) Bleaching correction: the fluorescent dye bleached slowly during the exposure to light, and therefore, we subtracted from each frame an exponential decay curve that was estimated from the bleaching course of frames 3–7 and frames 26–40 (i.e., before and after stimulus and response). (3) Median filtering: a spatial median filter with a width of 7 pixels was applied to remove outliers. (4) Movement correction: possible shifts of the antennal lobe from one stimulation experiment to the next one were corrected by aligning frame 20 of each experiment to frame 20 of the median experiment in a given animal. The outline of the antennal lobe and remains of tracheae served as guides for this movement correction procedure. Increased neural activity, indicated as an increase of the intracellular calcium concentration after stimulation with the diagnostic odors, was leading to spatially restricted spots of increased fluorescence in the antennal lobe. In the center of each activity spot, the average deltaF/F was recorded in an area of the size of a small- to medium-sized glomerulus (60 µm × 60 µm). Time traces of deltaF/F were averaged over three successive frames for each activity spot. In these smoothed time traces, the maximum deltaF/F after stimulus onset was determined. The average of the maximum value and the value before and after the maximum were calculated and were defined as the response of the animal to the odor stimulation at the given activity spot.

## Analysis of response to headspace stimulation

Activation patterns evoked by the diagnostic odors helped to establish an individual schematic of 23 putative glomeruli for each animal (*Figure 3*). Then, responses in these 23 glomeruli were calculated for stimulations with headspaces and dichloromethane. To identify headspace-activated glomeruli, we tested for each glomerulus if its mean response towards the two control stimulations with dichloromethane was clearly different from the response to the headspaces (p<0.01, Friedman test

with Dunn's multiple-comparisons test). We then normalized responses for each glomerulus in a given animal according to its response to headspaces and dichloromethane (lowest response = 0, highest response = 100) to balance for variability between individuals.

## Statistical analysis

Sample sizes and statistical tests used are given in the text and figure legends. Statistical tests were performed with PAST (version 3.26, http://folk.uio.no/ohammer/past/) and GraphPad InStat (version 3.10, GraphPad Software, San Diego, CA, https://www.graphpad.com).

## Acknowledgements

We thank Brett C Blum and Mark E Heitlinger for kindly hosting us at the Santa Rita Experimental Range, Sascha Bucks for breeding *M. sexta*, Kerstin Weniger for help with chemical analyses, Mohammed Khallaf Ali for introducing us to XCMS, and Daniel Veit for building equipment for mobile headspace collection. This work was funded by Max-Planck Society (all authors) and CSIRO Health and Biosecurity Acorn Grant (MAR).

## Additional information

### Funding

No external funding was received for this work.

### Author contributions

Sonja Bisch-Knaden, Conceptualization, Data curation, Formal analysis, Investigation, Methodology, Project administration, Supervision, Validation, Visualization, Writing - original draft, Writing - review and editing; Michelle A Rafter, Formal analysis, Funding acquisition, Investigation, Writing - review and editing; Markus Knaden, Conceptualization, Investigation, Writing - review and editing; Bill S Hansson, Conceptualization, Funding acquisition, Supervision, Writing - review and editing

### Author ORCIDs

Sonja Bisch-Knaden http://orcid.org/0000-0002-3424-8376
Michelle A Rafter http://orcid.org/0000-0002-5180-0062
Markus Knaden http://orcid.org/0000-0002-6710-1071
Bill S Hansson http://orcid.org/0000-0002-4811-1223

### Decision letter and Author response

Decision letter https://doi.org/10.7554/eLife.77429.sa1
Author response https://doi.org/10.7554/eLife.77429.sa2

## Additional files

### Supplementary files

• Transparent reporting form

### Data availability

Figure 1—source data 1, Figure 2—source data 1 and Figure 3—source data 1 contain the numerical data used to generate the figures. Figure 3—source code 1 contains custom written software for processing calcium imaging data.

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
