## [Editor Report]

This article is of particular interest to researchers in the fields of neuroecology of insect olfaction and of insect–plant interactions in general. The authors investigate the olfactory signals that guide the specialist hawkmoth *Manduca sexta* towards plants that are used for oviposition and for nectar-feeding in a natural setting. How insects distinguish useful information from irrelevant information is an important question. The authors use elegant chemical ecology techniques and recordings of neuronal activity to ask how female moths (*M. sexta*) could discriminate co-occurring behaviorally relevant versus irrelevant plant and floral volatiles.

---

## [Decision Letter]

**Decision letter after peer review:**

[Editors’ note: the authors submitted for reconsideration following the decision after peer review. What follows is the decision letter after the first round of review.]

Thank you for submitting the paper "Unique neural coding of crucial *versus* irrelevant plant odors in a hawkmoth" for consideration by *eLife*. Your article has been reviewed by 3 peer reviewers, and the evaluation has been overseen by a Reviewing Editor and a Senior Editor. The following individual involved in the review of your submission has agreed to reveal their identity: Sylvia Anton (Reviewer #3).

Comments to the Authors:

We are sorry to say that, after consultation with the reviewers, we have decided that this work will not be considered further for publication by *eLife*.

Your manuscript addresses the olfactory signals that guide the specialist hawkmoth Manduca sexta towards plants that are used for oviposition and for nectar feeding in a natural environment in Southern Arizona, US. The study describes how different complex, ecologically-relevant olfactory signals are detected and represented in the moth brain. Its interest derives from the fact that insects usually live in a complex odorant space, which contains myriad odorants. How insects distinguish useful information from irrelevant information is an important question.

While being well done, the reviewers found the innovation of the study limited, as exemplified by a range of published papers; and they also raised the need to connect behavioural data and neurobiological data. In sum, the reviewers see the potential of this work to contribute to our understanding of how moths use plant volatiles to seek nectar and select host plants, but the agreement was that previous relevant work is not sufficiently recognized and reported, and that additional experimental work is needed to provide a substantial advance for the field.

*Reviewer #1 (Recommendations for the authors):*

General:

As someone interested in chemical ecology and insect-host naturalistic interactions, I highly appreciate and welcome this type of work. The use of natural systems and state of the art methodologies and well-conceived and conducted experiments that couple signals with neuron activity is highly suited for better understanding how behavioral decisions, such as feeding and choosing an appropriate oviposition site, are governed. Technically and intellectually, the experiments are well conducted and the amount of collected information is impressive. The manuscript, however, somehow lacks in novelty because much of it has been previously reported, in particular, the chemical composition and bioactivity of components in the floral odor of *D. wrightii* and *A. palmieri*, and how those two nectar sources differentially activate antennal lobe neurons (Riffell et al. 2008, 2009a, 2009b, 2014). These previous results highly overlap with the findings presented here. How moths could identify a behaviorally relevant odor bouquet (such as the floral scent of *D. wrightii*) in an environment of irrelevant odors (e.g. creosote bush) has also been previously examined (Riffell et al. 2014), and it would be good to discuss this here.

*D. wrightii* is used both as a nectar source for adults and hostplant for the larvae, but previous work showed that the presence of flowers increases oviposition both in the field and in laboratory experiments (Reisenman et al. 2010), suggesting that floral odors (owing to the high emission of VOCs) can attract females at a distance for oviposition. In particular, a compound within the *D. wrightii* flower odor (linalool, which is typical of highly reflective white night-blooming flowers) selectively activate PNs in a female specific glomerulus, providing further support for this idea.

The imaging experiments are impressive, in an insect for which neuronal markers that could facilitate this are not available. However, the conclusions from this experiment only apply to those glomeruli (one third approximately) imaged. This is mentioned throughout the manuscript, but the authors should be careful in their statements (e.g. lines 240-242, 247-252, 452-453, 471-473)

Introduction:

Lines 45-46: In the case of Manduca sexta in the environment described (Southern Arizona, USA), *D. wrightii* floral odors likely serve to additionally signal moths the presence of an oviposition site (Reisenman et al. 2010).

Lines 51-66: Previous work has characterized the components, amounts, and ratios within the natural flowers (same species as this work) used by *M. sexta* that are capable of mimicking the behavior towards the real flower. The neural activation patterns evoked by these sources at the antennal lobe level have been specifically examined in *M. sexta*, and the compounds that evoke strong responses identified. (Riffell et al. 2008, 2009a, 2009b, 2013). The influence of background odors (e.g. background vegetation from non-hosts) on the AL representation of behaviorally relevant blends has been examined as well (Riffell et al. 2014).

Lines 79-81: This sentence needs the following citations: Bronstein et al. (2009), Mechaber et al. (2002).

Line 80-81: The following reference could be added here, which discusses the idea of antagonistic mutualism for the system under study: Adler and Bronstein 2004.

Line 93-94: Because about one third of all glomeruli are imaged, it can't be discarded that further responses/discrimination take place in glomeruli not imaged. This should be mentioned whenever relevant, and statements should be cautious (e.g. lines 452-453: "m sexta hostplant bouquets each activated only a single glomerulus"; but see also 240-242, 247-252, 471-473)

Line 96-97: Learning is an important factor that modulates the representation of odors in the AL (Daly et al. 2004, Riffell et al. 2013).

Lines 103-104: This finding is in agreement with previous findings, strong responses to odors related to nectar sources (Riffell et al. 2009a,b). Although odors from vegetative parts of hosts elicit weak responses, in the case of *D. wrightii* moths likely use the floral odors to signal oviposition sites. Linalool, which is present in the floral scent of *D. wrightii* (but not in *A. palmieri*) strongly activates a female-specific olfactory glomeruli (King et al. 2000). Evidence suggests that these glomeruli are necessary and sufficient to mediate oviposition (Kalberer et al. 2010). Vegetative host-plant odors (tomato leafs) have been also reported to evoke responses from the two female-specific glomeruli (King et al. 2000, Reisenman et al. 2009).

Results and discussion:

In general, the findings regarding the number of GC peaks, the chemical composition and identity need to be discussed in the context of previous findings because there's substantial overlap. Previous work identified some of the same chemicals as most abundant and strong activators (e.g. ethyl sorbate in Agave flower, benzyl alcohol, ocimene, geraniol, linalool in Datura flowers, Figure 2C). The host-plant vegetative odors of intact and larva-damaged natural hosts *D. wrightii* and *D. discolor*, and cultivated tomato have been investigated previously and should be cited here (Reisenman et al. 2013).

Figure 1: The GC traces in Figure A is not very informative because the reader does not know what the different peaks are. If the purpose is to show that there are a lot more peaks (more components) in the floral nectar, or in host sympatric hosts, that's also achieved in Figure B. Maybe provide a key to some of the components for each sample?

Lines 157-171: Refer to previous findings about those same compounds from at least two of the sources being strong activators (e.g. aliphatic esters, terpenes and aromatics, Figure 2C). Although many components are present, reduced floral mimics with just 3-6 compounds were shown to be sufficient to evoke behavior identical to that evoked by real flower, at least in the laboratory setting (Riffell et al. 2009a,b).

Lines 166-177: It should be mentioned here that projections neurons in a sexually isomorphic glomerulus in *M. sexta* are selective and highly sensitive (below 10-6 vol/vol dilution in females) to cis-3-hexenyl-acetate (Reisenman et al. 2005).

Line 240: This statement applies only to the glomeruli imaged here (ca. 1/3 of all AL glomeruli) and so this should be acknowledged.

Lines 344-348: It should also be cited a previous study showing high specificity and sensitivity towards cis-3-hexenyl-acetate in projection neurons from an identified glomerulus (Reisenman et al. 2005). These PNs respond most strongly to that compound, but also respond to another ester, cis-3-hexenyl propionate.

Lines 355-357: Strong responses to Agave floral odors previously reported (Riffell et al. 2009b)

Lines 360-361: A clear distinction should be made between neuron responses and behavioral responses when discussing learning.

Lines 364-366: Indeed there are PNs which are selective and extremely sensitive to this odor compound (below 10-6 vol/vol) in a sexually isomorphic glomerulus (Reisenman et al. 2005).

Lines 399-405: Other reports in M sexta show widespread inhibition in the AL (Lei et al. 2004) and that altering the balance of excitation and inhibition alters blend odor representation impeding tracking (Riffell et al. 2014).

Lines 444-451: Previous findings suggest that these glomeruli are involved in mediating oviposition, at least in part: (1) At least some PNs in these glomeruli (ca. 20%) respond to vegetative odors from tomato (a plant used for only oviposition) (King et al. 2000, Figure 7), and some examples were also reported for the medLFG (Reisenman et al. 2009, Figure 1). (2) Experiments in which the antennal imaginal disk of a female is transplanted in the developing male larvae show that the presence of the induced female glomeruli in these gynandromorph animals is necessary and sufficient for orientation towards host-plants (Kalberer et al. 2010).

It is also possible that the LFGs use floral odors to orient females towards oviposition sites (in the case of datura at least) because: 1) the latLFG (glomerulus # 2 in Figure 3A) is activated by Datura floral odorants (Figure 3B), responds selectively to (+)-linalool (Reisenman et al. 2004, Bisch-Knaden 2018), an odorant which occur in hawkmoth pollinated flowers including *D. wrightii*, Raguso and Pichersky 1999; Reisenman et al. 2010); 2) females oviposit much more in presence of Datura flowers, including vegetation presenting a mimic floral scent containing (+) but not (-) linalool (Reisenman et al. 2010); responses to the enantiomers being dependent on context and accession (He et al. 2019); 3) the female antenna expresses one female-specific OR which is homologous of a Bombyx mori female-specific OR which detects linalool (Grobe-Wilde et al. 2011, Anderson et al. 2008).

Lines 510-513: It has been shown that *M. sexta* has reduced oviposition in some larva-damaged hosts and that total emission of VOCs are higher in these plants (Reisenman et al. 2013).

Lines 531-540: The effect of background odors in odor tracking of relevant source has been investigated to some extent in the *Datura wrightii/M. sexta* system (Riffell et al. 2014), showing that moths can track the source better in backgrounds of non-overlapping odorants.

*Reviewer #2 (Recommendations for the authors):*

1. This work is based primarily on *M. sexta*-plant relationships previously reported and does not include behavioral experimental data. It would be very helpful to explain the present results if some behavioral data are available.

2. In the headspace collections, the flowers or branches were used for Agave palmeri and Datura wrightii. Why the authors did not use the flower branches, which should be comparable with the treatments for other plants? From the present data, the flower had a great influence on odorant collection.

3. How about the male moth's responses to these odors? Generally, both male and female moths search for nectar, while only female moths search for oviposition sites. By analyzing how male and female moths respond to these scents, it may be possible to estimate which scents are associated with the nectar searching or oviposition searching.

4. In Figure 2A, the GC-EAD recordings of Datura wrightii was not included, why? I think Datura should be one of the most important plants in the system.

5. Overall, the in vivo calcium imaging experiment is not complete. It is better to link the active glomeruli with some important EAD-active compounds. Are the compounds with high EAD activities the ones that activate strongly the glomeruli?

6. The activation of glomeruli depends on mating status. Is this difference derived from the differential peripheral inputs or the changed modulation on the activity of glomeruli by pre-and postsynaptic modulation?

7. I suggest that the odorants from two hostplants activating the glomeruli in the antenna lobe be identified. It would be very nice if they can also determine the attractiveness of these chemicals to the mated adults.

*Reviewer #3 (Recommendations for the authors):*

This is an easy to read, very exciting manuscript. I only have a few minor suggestions for improvement.

In Figure 2C at least in the version I have, some numbers are cropped on the right side (those >10).

Line 272: graph depicts

Line 677: replace "fluorescent" either by "fluorescent dye" or "fluorescence"?

Discussion lines 310-315:

When you discuss the antennal detection of volatiles, you talk about discrimination capability. I would maybe not talk about discrimination at this level. You find indeed differential detection, which could provide them with the necessary information to discriminate, but I would maybe formulate this with a little more caution.

In the same paragraph, I am not sure that I can follow your argument that detection and discrimination capability appeared to be better than could be inferred from the chemical analysis.

[Editors’ note: further revisions were suggested prior to acceptance, as described below.]

Thank you for resubmitting your work entitled "Unique neural coding of crucial *versus* irrelevant plant odors in a hawkmoth" for further consideration by *eLife*. Your revised article has been evaluated by Meredith Schuman (Senior Editor) and a Reviewing Editor.

The manuscript has been improved but there are some remaining issues that need to be addressed, as outlined below:

The three reviewers are supportive of the revised manuscript and the revisions you made as well as the extensive explanations you provided in the rebuttal. The last suggestions for modification, made by reviewers #1 and #2 involve some explanation to support some statements that will further increase the quality of the manuscript.

*Reviewer #1 (Recommendations for the authors):*

I thank the authors for taking the time to carefully consider the reviewers comments, making suggested changes, clarifying some of their statements, etc. The addition of sentences clarifying differences and similarities with previous studies, both methodological and in terms of findings, is very helpful and useful for a reader who is not particularly familiar with the specifics of the system under study.

With the changes introduced by the authors, I think that the manuscript now better conveys the novelty and relevance of the findings: while much was previously known about some of the important sources used by hawkmoths for nectar feeding and oviposition (e.g. Datura and Agave flowers, Datura and Prosbocidea foliage), the comparison in terms of chemistry and antennal and antennal lobe responses with those of non-host plant provides a framework for better explaining how moths can distinguish relevant from irrelevant hostplants and nectar sources.

I think that in its present form, this is a very interesting study about how specialist insects find olfactory important resources, distinguishing from those that are irrelevant or non-suitable, in their environmentally complex olfactory environments. The combination of field collections in the moths' natural environmental, and of chemical analysis coupled to recording from peripheral olfactory organs and imaging of neuronal activity in the primary olfactory center, plus comparison between virgin and mated females, is a strength of the manuscript.

One general comment that I have is regarding the comparison between antennal responses to single (GC-EAD) compounds and antennal lobe responses (vegetative/floral blends), and comparison with previous studies which use GC-multiunit recording. It is not surprising to me that the full blend evokes responses in many glomeruli, as the responses at this level might also be due to interglomerular interactions (reciprocal synapses, inhibition, etc.): it is possible that single compounds evoke responses in few glomeruli, while blends evoke wide-spread responses. I think it is a good addition that the authors now describe the difference between methodologies and what can be learned from each of them. There's also a couple of points (in the list below) that I think the authors should revise (comments on lines 461-471, 502-503, 571-573).

Line 20: "Responses to bouquets"

Line 77: the sentence starting with "However.." should be directly after the previous one for better flow of the logic.

Line 263: "…mating status (Table 2), at least among the 23 glomeruli imaged in this study."

Line 264: … activated only one glomerulus out of the 23 imaged in the antennal lobe…."

Regarding Figure 3C: which plant headspace is used here? This needs clarification. I assume is a single species headspace -it wouldn't be appropriate to mix headspaces (the captions says "plant headspaces").

Line 345: between the bouquets of host plant vegetation and surrounding…"

Line 351: "..in typical hawkmoth-pollinated flowers…"

Line 359: "…. Is at least as sensitive to promising floral blend…"

Line 367: "…with the duration a female shows feeding behavior (i.e proboscis contact time with a scented filter paper flower, Pearson correlation…., Bisch-Knaden et al. 2018). In contrast…"

Line 371: "… i.e. a behavior related to oviposition….."

Line 381: "In addition, the antenna might harbor narrowly tuned olfactory receptor neurons strongly responding…"

Line 414: "… the active GC-peaks overlapped between…"

Line 424: "…influence both the composition and the quantity…"

Line 444: "… presynaptic level, i.e. at the level…"

Line 450: "…. not elicit activity in the 23 imaged antennal lobe glomeruli."

Lines 461-471: I think the explanation for the different studies misses the fact that while single components might not evoke broad activity at the AL lobe level, blends/mixtures might do so due to emergent properties of AL circuitry. The current study uses blends as stimuli for imaging of AL activity, which might explain that the authors found broad activation across the array of imaged glomeruli. So the two studies not only used different techniques, each with its own advantages, but seek to answer different questions. Indeed, in their previous publication (Bisch-Knaden et al. 2018), they used monomolecular odorants and for the most part each odorant activates a few glomeruli (at least medium to strong, Figure 2D), including the esters. In line 469 the authors say that ethyl sorbate and benzyl alcohol evoke responses in the AL in the previous study; the responses to ethyl sorbate are small and limited mostly to glomeruli 12 and a few others; similarly, the responses to benzyl alcohol were not very strong and involved about 5 glomeruli. In their previous study, monoterpenes seem to evoke the strongest responses and more widespread (i.e. involving more glomeruli). The way the ms refers to these results gives the reader the impression that the single monomolecular odorants evoke broad responses, comparable to the blend-evoked responses, which I don't think is correct. I suggest that the authors modify this paragraph accordingly.

Line 500: "… to evoke a response in sensilla targeting mostly (but not only) the two female specific glomeruli" (Shields & Hildebrand shows that while most sensilla dye-filled target the LFGs, some target a few other glomeruli).

Lines 502-503: I still think that the authors do not have sufficient arguments for the statement at it is in this sentence. This is because: 1) the authors do not find response to vegetation in the LFGs, but the imaging technique, as the authors state, reveals activity from AL afferent mostly, not AL outputs. Although a cultivated plant, King et al. (2000) and Reisenman et al. (2009) reported conspicuous responses to tomato leaves in one of these glomeruli; 2) it is possible that the LFGs act in concert with other glomeruli to guide oviposition behavior (concerted responses not revealed by imaging of afferents might have important downstream effects); 3) males with induced LFGs fly more towards host plants (because these are the most prominent female specific glomeruli, this suggest that these glomeruli process some odorants which directly or indirectly signal an oviposition site. I thus suggest for this line something like this: "In spite of this, it is still possible that these female-specific glomeruli act in concert with other glomeruli to guide the female-specific behavior of identifying an oviposition site, a hypothesis that need further investigation."

Line 520: "However, host plants activate only one of these glomeruli, …… activated additional glomeruli. While it is possible that host plants activate glomeruli not imaged in this study, the resulting neural representation…"

Line 550: here add Goyret 2010 (J Exp Biol, Look and touch: multimodal sensory control of flower inspection movements in the nocturnal hawkmoth Manduca sexta).

Line 552: "…when searching for oviposition…"

Line 571-573: I still think that the case of *D. wrightii* is particularly interesting and the fact that the plant has a faint vegetative scent but powerful floral odor has significance. It is entirely possible and supported by previous findings that female might simply use the floral odors for long distance olfactory attraction, as these are fragrant and abundant (Raguso et al. 2003, and evoke strong responses). Once in the vicinity or closer, females might use vegetative odors to decided whether or not oviposit on leaf tissues (in addition to feeding on nectar, as it is known that females mix feeding and oviposition bouts). Therefore, the two processes, oviposition and feeding, guided by weak and strong odors but at different timescales, might entangled with each other in the *M. sexta*-*D. wrightii* system. Also, it is commonly observed that *M. sexta* moths mix oviposition and feeding bouts on this plant, and it is reported that flowers increased oviposition both in the lab and in the field (Reisenman et al. 2010). I suggest the authors modify the sentence starting with "Hence, *M. sexta*…" to reflect this fact.

Line 583: "… in contrast to the weak but specific activation of single glomeruli (among those imaged) by host plants of *M. sexta*."

*Reviewer #2 (Recommendations for the authors):*

The authors responded to the questions I raised, and then I think that this work deserves publication in the journal. However, I still have two suggestions:

(1) Since Datura wrightii is a unique and important plant in this study system, which is both the nectar source and host plant of this moth species, I'm sticking to my opinion that the representative traces of GC-EAD and in vivo calcium imaging recordings of headspace stimulations of the flowers and foliage of Datura wrightii should be added in Figure 2A and Figure 3B although these data were reflected in other figures and source data. For floral and foliage odors of Datura could attract females for oviposition, a comparison of responses of antennae or antennal lobes between floral and foliage of the same plant would be significant.

(2) The EAD activities and calcium imaging activities are the core contents of this study, it is better to analyze their linkage. As the dye used in vivo calcium imaging experiments is Calcium Green-1 AM, the information reflected in the calcium imaging activity may be mainly the input of the olfactory sensory neurons from the antennae. Therefore, there should be positive relationships between EAD activities and calcium imaging activities in theory. I am very curious about the following questions: based on the authors' previous research (Bisch-Knaden et al., 2018), which compound(s) could elicit activities of the only glomerulus (Glomerulus 4) activated by Datura foliage? Were these compounds present in the headspace blend of Datura foliage? If so, how about EAD responses or even behavioral responses to these compounds? The same questions with Glomerulus 15 and 12 for Proboscidea. These could be discussed in the Discussion section.

---

## [Author Response]

[Editors’ note: The authors appealed the original decision. What follows is the authors’ response to the first round of review.]

Reviewer #1 (Recommendations for the authors):General:1. As someone interested in chemical ecology and insect-host naturalistic interactions, I highly appreciate and welcome this type of work. The use of natural systems and state of the art methodologies and well-conceived and conducted experiments that couple signals with neuron activity is highly suited for better understanding how behavioral decisions, such as feeding and choosing an appropriate oviposition site, are governed. Technically and intellectually, the experiments are well conducted and the amount of collected information is impressive.

We thank the reviewer for this positive evaluation of our study in general.

2. The manuscript, however, somehow lacks in novelty because much of it has been previously reported,

We are surprised about the statement that our study lacks novelty.

A) We collected nocturnal headspace in the field (17 samples from 16 different plant species); 13 of these 17 headspaces (76% new data) have not been published before to the best of our knowledge. The previously published headspaces are floral scents of Agave and Datura (see our reply to point 3), and leaf scents of Datura and Proboscidea (see B).

B) We next tested the antenna of female moths in GC-EAD experiments. Of the species tested here, only Datura foliage and Proboscidea have been studied before by Fraser et al. 2003; a study we mention in the introduction (line 58) and the discussion in detail (line 374-390). In Riffell et al. 2014, a GC-EAD example trace for stimulation of a male antenna with Datura headspace is shown in the supplementary material. However, no detailed analysis is presented except two arrowheads marking EAD-responses towards unspecified oxygenated aromatic compounds. Hence, GC-EAD results from 15 out of 17 tested headspaces are new data (88%).

C) Regarding headspace-evoked activation of identified glomeruli in the female antennal lobe, we are aware of two studies showing results for one single glomerulus in each. King et al. 2000 recorded intracellularly from projection neurons innervating one identified glomerulus using a tomato leaf as stimulus (line 445). Reisenman et al. 2009 recorded from another identified glomerulus using also a tomato leaf. We now added this reference based on the reviewer’s comment (see our reply to point 18). However, headspace-evoked spatial activation patterns in an array of identified glomeruli (n=23) in the female antennal lobe have not been investigated before in *M. sexta* (100% new data; tomato was not included in our study as it is a crop species not present in the natural habitat under investigation). Furthermore, we studied for the first time the effect of the females’ mating status on detection at the antenna and for spatial coding in the antennal lobe.

3. … in particular, the chemical composition and bioactivity of components in the floral odor of *D. wrightii* and *A. palmieri*, and how those two nectar sources differentially activate antennal lobe neurons (Riffell et al. 2008, 2009a, 2009b, 2014). These previous results highly overlap with the findings presented here.

Raguso et al. 2003 and Raguso 2004 first described the chemical composition of floral odors of *D. wrightii* and *A. palmeri*. In our original manuscript, we cited Raguso et al. 2003 for the scent of Datura, but unfortunately missed citing Raguso 2004 for the scent of Agave. We now added Raguso 2004, and based on the reviewer’s suggestion, we added as well Riffell et al. 2008 as an additional reference for the scent of these two flowers (revised line 88).

Regarding the chemical composition, we appreciate that our results for Agave and Datura overlap with the former studies and now mention this in the discussion (revised line 323-326). “Our results regarding the number and identity of compounds emitted by *Agave* and *Datura* flowers, and the dissimilarity between both floral bouquets confirm earlier studies (Raguso 2004; Raguso et al. 2003; Riffell et al. 2008).”

Regarding the bioactivity of components in the same two flowers that Riffell et al. studied in the male antennal lobe, we do not agree with the reviewer’s opinion as there is only little overlap with our results. Riffell et al. 2009 report that the neurons they recorded from responded only to a small fraction (Datura: 9 compounds, Agave: 6 compounds) of the > 60 compounds present in each of the headspaces. We found in our GC-EAD recordings that more fractions were active at the level of the antenna (Datura: 17 compounds, Agave: 20 compounds), and in our calcium imaging recordings, we found that both floral bouquets activated each of the 23 identified glomeruli in the antennal lobe.

One reason for the discrepancy between Riffell et al. and our study might be that Riffell et al. used male moths while we worked with females. Probably most important, however, are the different recording methods used by the labs. We briefly discussed this topic in a previous paper (Bisch-Knaden et al. 2018), but realize that it might be necessary to discuss it here again and in more detail as the reviewer and many readers might not be aware of the methodological differences, which have significant effects on the results and conclusions. Riffell et al. investigated antennal lobe activation in male moths using a multiunit neural-ensemble probe with 4 recording shanks in a linear row, with a distance of 125 μm from each other and a recording depth of 200 μm, i.e. about half the depth of the antennal lobe. The first shank is inserted in the macroglomerular complex as a reference point and not analyzed further, the other 3 shanks are inserted parallel to the axis of the antennal lobe with each shank impaling up to 3 glomeruli in the lateral region of the lobe, i.e. less than 10 glomeruli are impaled in each animal (Riffell et al. 2009). From experiment to experiment, the targeted glomeruli are adjacent but not always identical due to the impossibility to place the recording probe exactly in the same position and anatomical variability between animals. This recording method therefore allows analyzing the temporal characteristics of odor-evoked activity in the respective region (several glomeruli and some part of the central neuropil) of the male antennal lobe. However, it does not allow conclusions regarding the activation patterns across identified glomeruli; a limitation that is stated by Riffell and colleagues in all of their publications: “Although identification of individual glomeruli is an important prerequisite for assigning functional significance to a given glomerulus, it was beyond the scope of this study.” This is, in contrast, the strength of our calcium imaging study as we could consistently address the same 23 glomeruli in the dorsal part of the antennal lobe by testing a panel of diagnostic odorants in each animal based on our previous work (Bisch-Knaden et al. 2018), followed by stimulation with plant headspaces. We were thus able to assign functional significance to a given glomerulus and analyze the spatial representation of the tested headspaces across the same 23 glomeruli in the female antennal lobe. On the other hand, we did not analyze temporal coding patterns because calcium signals are rather slow, and as calcium influx indicates neural activation, no clear conclusions about inhibitory networks can be drawn. The analysis of temporal characteristics of odor-evoked activation of mixed populations of local interneurons and projection neurons in the antennal lobe, and the study of inhibitory interactions between these neurons, are the strength and advantage of the multiunit neural-ensemble recording method used by the Riffell lab. We therefore added a paragraph in the introduction explaining the different methods (revised line 59-72).

“At the level of the first olfactory processing center in the insect brain, the antennal lobe, previous studies have investigated the temporal coding patterns evoked by floral bouquets. By impaling a restricted region of the antennal lobe with a multiunit probe, simultaneous recordings from a mixed population of local interneurons and projection neurons in this area were performed (Riffell et al. 2009a+b). Using the same approach, inhibitory interactions between these neurons could be studied in addition (Lei et al. 2004). However, this recording technique does not allow assigning functional significance to individual olfactory glomeruli, which are the morphological and functional subunits of the antennal lobe (Gao et al. 2000; Hansson et al. 1992). An analysis of the spatial coding patterns, however, is possible *via* functional calcium imaging. Although this technique does not inform about temporal coding patterns or inhibitory interactions, it was used in different insect species to provide a detailed insight into the spatial representation of natural odor blends across the glomerular array.”

4 How moths could identify a behaviorally relevant odor bouquet (such as the floral scent of *D. wrightii*) in an environment of irrelevant odors (e.g. creosote bush) has also been previously examined (Riffell et al. 2014), and it would be good to discuss this here.

We apologize that we missed mentioning this paper in the original version of the manuscript. The findings by Riffell et al. 2014 are indeed interesting; however, they seem contradictory to the fact that creosote bushes (the only background species studied in Riffell et al. 2014) are obviously omnipresent in some Datura/Manduca environments, while Datura still regularly becomes pollinated by the moth in the same habitat (Alarcon et al. 2008). We did not observe creosote bushes in the near vicinity of the Datura plants in our restricted study area, at the time (August 2018) when we collected headspaces. We therefore could not include creosote in our collection of eleven background plant species. Anyhow, we now added the study of Riffell et al. 2014 in the discussion (revised line 463-465). “Interestingly, lab experiments in a wind tunnel suggest that even such a strong scent as the one from *Datura* flowers can become less attractive to male *M. sexta* when presented in the olfactory background of a selected non-host plant (Riffell et al. 2014).”

5. *D. wrightii* is used both as a nectar source for adults and hostplant for the larvae, but previous work showed that the presence of flowers increases oviposition both in the field and in laboratory experiments (Reisenman et al. 2010), suggesting that floral odors (owing to the high emission of VOCs) can attract females at a distance for oviposition.

We agree that this is interesting to mention and added the reference to a sentence in the introduction (revised line 94-98). “*Datura* plants thus have to interact with an insect that is at the same time an important pollinator and a damaging herbivore (Bronstein et al. 2009), enabling the moth to find an oviposition site by navigating towards the scent of nectar-providing flowers emitted by the same plant (Reisenman et al. 2010).”

However, as Datura foliage alone can attract egg-laying *M. sexta*, we decided to analyze bouquets from foliage and flowers separately and added an explanation to the method section (revised line 606-609). “Although flowering *Datura* plants receive more eggs than non-flowering *Datura* plants (Reisenman et al. 2010), foliage alone attracts egg-laying females in the field (Allmann et al. 2013) and in the lab (Spaethe et al. 2013). We therefore tested flowers and leaves of *Datura* separately.”

6. In particular, a compound within the *D. wrightii* flower odor (linalool, which is typical of highly reflective white night-blooming flowers) selectively activate PNs in a female specific glomerulus, providing further support for this idea.

Linalool is a very common volatile organic compound, not only a floral odor. Linalool also acts for example as indirect plant defense against herbivores; therefore, linalool is one of the compounds that can repel ovipositing moths (Kessler and Baldwin 2001). In our study, linalool was not only emitted from Datura flowers but in addition from non-flowering plants at a concentration that was EAD-active (Figure 2C). Finally, several glomeruli besides the female-specific LFGs respond to linalool (Hansson et al. 2003, Bisch-Knaden et al. 2018, Reisenman et al. 2004). In our opinion, discussing a single odorant with many and sometimes opposing potential meanings for *M. sexta* would distract the reader from the topic of our manuscript.

7. The imaging experiments are impressive, in an insect for which neuronal markers that could facilitate this are not available. However, the conclusions from this experiment only apply to those glomeruli (one third approximately) imaged. This is mentioned throughout the manuscript, but the authors should be careful in their statements (e.g. lines 240-242, 247-252, 452-453, 471-473)

We indeed mention this throughout the manuscript: results line 223-239, Figure 3, Table 2, methods line 694-6. Based on the reviewer’s suggestion we added this information again when we discuss the sparse representation of host plant and background odors.

“In contrast to the wide and strong activation of antennal lobe glomeruli by flower odors, *M. sexta*’s host plant bouquets each activated only a single glomerulus of the 23 glomeruli under investigation.” (revised line 496-497).

“Interestingly, virgin and mated females differed markedly in their response to the odor of background plants: out of the 23 identified glomeruli, these plants activated only a small number in virgin females (range: 0-9), and even less glomeruli (range: 0-2) in mated females.” (revised line 515-517).

Introduction:8. Lines 45-46: In the case of Manduca sexta in the environment described (Southern Arizona, USA), *D. wrightii* floral odors likely serve to additionally signal moths the presence of an oviposition site (Reisenman et al. 2010).

Please see our reply to point 5.

9. Lines 51-66: Previous work has characterized the components, amounts, and ratios within the natural flowers (same species as this work) used by *M. sexta* that are capable of mimicking the behavior towards the real flower.

This comment is about the floral scents of Agave and Datura, and their reduced mimics that were described as being as attractive to male *M. sexta* as the whole bouquets (Datura: 9 compounds in Riffell et al. 2009a, and 3 compounds in Riffell et al. 2009b; Agave: 6 compounds (Riffell et al. 2009b)). There was, however, no difference in the attractiveness of the main component of the published floral mimics and the whole mimic when females were tested in the wind tunnel (Bisch-Knaden et al. 2018). We already discussed in Bisch-Knaden et al. 2018 that this discrepancy might be based on the sex of the moths tested (Riffell used male moths), or on methodological differences. We, therefore, would prefer to not cite Riffell et al. 2009a+b in this context, as the discussion of the contradictory results would distract too much and is not relevant for the present study.

10. The neural activation patterns evoked by these sources at the antennal lobe level have been specifically examined in *M. sexta*, and the compounds that evoke strong responses identified. (Riffell et al. 2008, 2009a, 2009b, 2013).

Please see our reply to point 3.

We added a paragraph in the discussion dealing with this (revised line 453-463):

“In former studies, only a small fraction of the compounds present in the two floral bouquets (*Agave*: 10%, *Datura*: 15%) activated neurons in the antennal lobe of male *M. sexta* (Riffell et al. 2009a+b). However, the recording technique used by Riffell and colleagues targets about 10 glomeruli and part of the adjacent neuropil in the lateral part of the male antennal lobe, while we recorded from 23 identified glomeruli in the dorsal part of the female lobe. The differing results might thus be due to these different approaches or to potential sex-specific coding differences. However, compounds that were identified to activate neurons in the male antennal lobe, like ethyl sorbate (*Agave*) and benzyl alcohol (*Datura*), were also EAD-active in our present study (Figure 2C) and evoked responses in the female antennal lobe in a previous imaging study (Bisch-Knaden et al. 2018).”

11. The influence of background odors (e.g. background vegetation from non-hosts) on the AL representation of behaviorally relevant blends has been examined as well (Riffell et al. 2014).

Please see our reply to point 4.

12. Lines 79-81: This sentence needs the following citations: Bronstein et al. (2009), Mechaber et al. (2002).

We now cite Bronstein et al. 2009 as a reference for the mutualism between pollinating/damaging insects and plants. Mechaber et al. 2002 report the effect of age and mating status on the behavior of female *M. sexta* (we cited this paper in the original manuscript in another context, line 98). The reviewer probably meant Mechaber et al. 2000, which we cited in the original manuscript just one line afterwards (line 82).

13. Line 80-81: The following reference could be added here, which discusses the idea of antagonistic mutualism for the system under study: Adler and Bronstein 2004.

Adler and Bronstein 2004 deal with the question if the amount of floral nectar might affect larval herbivory in *Datura*. This topic seems to be beyond the scope of our manuscript.

14. Line 93-94: Because about one third of all glomeruli are imaged, it can't be discarded that further responses/discrimination take place in glomeruli not imaged. This should be mentioned whenever relevant, and statements should be cautious (e.g. lines 452-453: "m sexta hostplant bouquets each activated only a single glomerulus"; but see also 240-242, 247-252, 471-473)

Please see our reply to point 7.

15. Line 96-97: Learning is an important factor that modulates the representation of odors in the AL (Daly et al. 2004, Riffell et al. 2013).

We agree that learning is an important factor; however, we tested only naïve animals with no previous experience with plant odors or food sources. To clarify this point already in the introduction, we moved a sentence from the discussion to the introduction (revised line 115-117).

“The moths tested in our study were laboratory-reared on artificial diet, naïve to plant odors, not fed, and tested only once, as we were interested in the insets’ innate neuronal responses.”

16. Lines 103-104: This finding is in agreement with previous findings, strong responses to odors related to nectar sources (Riffell et al. 2009a,b).

The reviewer mentioned this before; please see our reply to points 3 and 10.

17. Although odors from vegetative parts of hosts elicit weak responses, in the case of *D. wrightii* moths likely use the floral odors to signal oviposition sites.

The reviewer mentioned this before; please see our reply to point 5.

18. Linalool, which is present in the floral scent of *D. wrightii* (but not in *A. palmieri*) strongly activates a female-specific olfactory glomeruli (King et al. 2000). Evidence suggests that these glomeruli are necessary and sufficient to mediate oviposition (Kalberer et al. 2010). Vegetative host-plant odors (tomato leafs) have been also reported to evoke responses from the two female-specific glomeruli (King et al. 2000, Reisenman et al. 2009).

We cited King et al. 2000 in the original manuscript as a reference for the assumption that LFGs might be involved in mediating oviposition (line 445), and now added Reisenman et al. 2009 as well (revised line 488). However, Reisenman et al. 2009 show an example tomato-response trace but give no sample size or analyzed data for this result. We therefore previously judged the tomato-result as anecdotal evidence. There is, however, substantial data presented in Shields and Hildebrand 2000, who found that vegetative headspace from tomato (13 sensilla tested), failed to evoke a response in sensilla targeting the two LFGs. In addition, no response was found for leaves of Datura (18 sensilla tested) and Nicotiana (3 sensilla tested). We cited this paper in our original manuscript as a reference for the assumption that LFGs might not be involved in mediating oviposition (line 449).

The gynandromorph males in Kalberer et al. 2010 developed a complete female antennal lobe; we see no evidence that only the LFGs might be responsible for the observed female-like attraction towards host plants. Besides the two LFGs, there are three more female-specific glomeruli (Grosse-Wilde et al. 2011). We are currently investigating the role of female-specific glomeruli and the corresponding receptors in another study.

Results and discussion:19. In general, the findings regarding the number of GC peaks, the chemical composition and identity need to be discussed in the context of previous findings because there's substantial overlap. Previous work identified some of the same chemicals as most abundant and strong activators (e.g. ethyl sorbate in Agave flower, benzyl alcohol, ocimene, geraniol, linalool in Datura flowers, Figure 2C). The host-plant vegetative odors of intact and larva-damaged natural hosts *D. wrightii* and *D. discolor*, and cultivated tomato have been investigated previously and should be cited here (Reisenman et al. 2013).

The reviewer mentioned this before; please see our reply to points 3 and 10.

Reisenman et al. 2013 investigated two Datura species and cultivated tomato. They show that oviposition was reduced in larva-damaged tomato plants but not in larva-damaged Datura plants, although all damaged plant species emitted more volatiles than intact plants. Tomato is a crop species not involved in our study, and in addition, we studied the headspace of natural plant populations with omnipresent herbivory. We therefore prefer not citing this reference here in order not to confuse the reader.

20. Figure 1: The GC traces in Figure A is not very informative because the reader does not know what the different peaks are. If the purpose is to show that there are a lot more peaks (more components) in the floral nectar, or in host sympatric hosts, that's also achieved in Figure B. Maybe provide a key to some of the components for each sample?

In Figure 1A, we show representative GC traces for each headspace to illustrate i) the abundance of compounds, analyzed in Figure 1B, and ii) the difference in chemical composition (GC-peaks at different retention times) and in the amount of emissions (GC-peaks of different heights), displayed in Figure 1C.

However, in the representative GC-EAD traces shown in Figure 2A, we provide a key for those GC-peaks that are EAD-active (shown in Figure 2C) as only the identity of biologically active compounds is interesting in our context. Therefore, we would like to keep the presentation as it is.

21. Lines 157-171: Refer to previous findings about those same compounds from at least two of the sources being strong activators (e.g. aliphatic esters, terpenes and aromatics, Figure 2C).

In this section of the results, we describe our GC-EAD results across all 17 plant headspace samples. Regarding previous results for floral odors of Agave and Datura, see our reply to point 3 and 10.

22. Although many components are present, reduced floral mimics with just 3-6 compounds were shown to be sufficient to evoke behavior identical to that evoked by real flower, at least in the laboratory setting (Riffell et al. 2009a,b).

The reviewer mentioned this before; please see our reply to point 9.

23. Lines 166-177: It should be mentioned here that projections neurons in a sexually isomorphic glomerulus in *M. sexta* are selective and highly sensitive (below 10-6 vol/vol dilution in females) to cis-3-hexenyl-acetate (Reisenman et al. 2005).

Thanks for this suggestion; we added the reference in the discussion (revised line 398-401).

“Interestingly, some projection neurons in the female antennal lobe targeting an identified glomerulus were reported to specifically respond to low concentrations of (Z)-3-hexenyl acetate (Reisenman et al. 2005). “

24. Line 240: This statement applies only to the glomeruli imaged here (ca. 1/3 of all AL glomeruli) and so this should be acknowledged.

The reviewer mentioned this before; please see our reply to point 7.

25. Lines 344-348: It should also be cited a previous study showing high specificity and sensitivity towards cis-3-hexenyl-acetate in projection neurons from an identified glomerulus (Reisenman et al. 2005). These PNs respond most strongly to that compound, but also respond to another ester, cis-3-hexenyl propionate.

The reviewer mentioned this before; please see our reply to point 23.

26. Lines 355-357: Strong responses to Agave floral odors previously reported (Riffell et al. 2009b)

Response to Agave odors are reported by Riffell et al. 2009b to be only moderate (only 6 out of >60 Agave components evoke a response in the neurons). Please see our reply to point 3 and 10.

27. Lines 360-361: A clear distinction should be made between neuron responses and behavioral responses when discussing learning.

We changed “responses” to “neuronal responses” and moved the sentence to the introduction (revised line 115-117) based on point 15.

28. Lines 364-366: Indeed there are PNs which are selective and extremely sensitive to this odor compound (below 10-6 vol/vol) in a sexually isomorphic glomerulus (Reisenman et al. 2005).

The reviewer mentioned this before; please see our reply to point 23.

29. Lines 399-405: Other reports in M sexta show widespread inhibition in the AL (Lei et al. 2004) and that altering the balance of excitation and inhibition alters blend odor representation impeding tracking (Riffell et al. 2014).

The reviewer mentioned this before; please see our reply to point 3.

30.1. Lines 444-451: Previous findings suggest that these glomeruli are involved in mediating oviposition, at least in part: (1) At least some PNs in these glomeruli (ca. 20%) respond to vegetative odors from tomato (a plant used for only oviposition) (King et al. 2000, Figure 7), and some examples were also reported for the medLFG (Reisenman et al. 2009, Figure 1). (2)

The reviewer mentioned this before; please see our reply to point 18.

30.2. Experiments in which the antennal imaginal disk of a female is transplanted in the developing male larvae show that the presence of the induced female glomeruli in these gynandromorph animals is necessary and sufficient for orientation towards host-plants (Kalberer et al. 2010).

The reviewer mentioned this before; please see our reply to point 18.

31. It is also possible that the LFGs use floral odors to orient females towards oviposition sites (in the case of datura at least) because:31.1. The latLFG (glomerulus # 2 in Figure 3A) is activated by Datura floral odorants (Figure 3B), responds selectively to (+)-linalool (Reisenman et al. 2004, Bisch-Knaden 2018), an odorant which occur in hawkmoth pollinated flowers including *D. wrightii*, Raguso and Pichersky 1999; Reisenman et al. 2010).

All 23 glomeruli tested in our study respond to Datura and Agave floral headspace (Figure 3, Table 2), and several glomeruli respond to linalool (Hansson et al. 2003, Bisch-Knaden et al. 2018, Reisenman et al. 20004). The finding in our previous imaging study (Bisch-Knaden et al. 2018) that glomerulus #2 responded stronger to (+)-linalool than to (-)-linalool was indeed a hint that glomerulus #2 might be the lateral LFG based on results from Reisenman et al. 2004. This enantiomer-selective response of the lateral LFG is definitely fascinating but we see little relevance to discuss this in our present work: i) Datura flowers emit a 50:50 mixture of linalool enantiomers (Reisenman et al. 2010), ii) Datura foliage emits no linalool (Reisenman et al. 2010 and our manuscript Figure 2C), iii) the enantiomeric ratio of linalool in the headspace of the host plant Proboscidea is unknown, and iv) linalool was an EAD-active compound emitted by a non-flowering host plant for sympatric hawkmoths and two background plants, one with flowers and one without flowers (Figure 2C), showing that linalool is not flower-specific. However, we feel that discussing this in detail is beyond the scope of this manuscript. Please see also our reply to point 6.

31.2. Females oviposit much more in presence of Datura flowers, including vegetation presenting a mimic floral scent containing (+) but not (-) linalool (Reisenman et al. 2010); responses to the enantiomers being dependent on context and accession (He et al. 2019);

Please see our comment to 5 and 31(1).

31.3. The female antenna expresses one female-specific OR which is homologous of a Bombyx mori female-specific OR which detects linalool (Grobe-Wilde et al. 2011, Anderson et al. 2008).

The expression of two female-specific ORs on the antenna of *M. sexta* that are homologous to a female-specific, linalool-detecting OR of B. mori (Grosse-Wilde et al. 2011, Anderson et al. 2009) is very interesting and implies that at least one of the two homologous *M. sexta* ORs might also detect linalool. We think this is a very plausible hypothesis; however, we see little relevance to discuss this single odorant (linalool) in our work dealing with the detection and coding of crucial versus irrelevant plant bouquets (see our arguments in response to points 6,18, 19, and 31(1). We briefly discuss only the three single odorants that were found to be extremely effective in activating the antenna although they were present in low concentrations (a-copaene, (Z)-3-hexenyl acetate, b-ocimene, Figure 2D), and the aliphatic esters present in Agave flower as they activated the antenna to a high degree (Figure 2D, E).

32. Lines 510-513: It has been shown that *M. sexta* has reduced oviposition in some larva-damaged hosts and that total emission of VOCs are higher in these plants (Reisenman et al. 2013).

The reviewer mentioned this before; please see our reply to point 19.

33. Lines 531-540: The effect of background odors in odor tracking of relevant source has been investigated to some extent in the *Datura wrightii/M. sexta* system (Riffell et al. 2014), showing that moths can track the source better in backgrounds of non-overlapping odorants.

The reviewer mentioned this before; please see our reply to point 4.

Reviewer #2 (Recommendations for the authors):1. This work is based primarily on *M. sexta*-plant relationships previously reported and does not include behavioral experimental data. It would be very helpful to explain the present results if some behavioral data are available.

The reviewer is right; our study is based on the well-documented ecological meaning of plant species that are nectar sources and host plants for *M. sexta* in its habitat in Arizona. This previously reported knowledge enabled us to decide which of the plants in the field are relevant for *M. sexta* and which are irrelevant. However, we did not aim at challenging previous results or showing again, which plants are relevant for *M. sexta* and which are not relevant by performing behavioral experiments. (see also our reply to the reviewer’s comment above.)

2. In the headspace collections, the flowers or branches were used for Agave palmeri and Datura wrightii. Why the authors did not use the flower branches, which should be comparable with the treatments for other plants? From the present data, the flower had a great influence on odorant collection.

We thank the reviewer for raising this question and now explain our choice in more detail in the method section (revised line 599-609) “We sampled from flowering plants or flowering branches if the respective plant was blooming during the experimental nights. Otherwise, non-flowering branches were sampled (Table 1). *Agave* is a succulent plant with a basal rosette of sharp-edged leaves, each with a length of c. 1m and long spines at the tip. These leaves did not fit in our collection bags (see below). We therefore only collected headspace from *Agave* flowers, which appear in umbels at the end of a long bloom stalk without leaves (about 5 to 6 m above the basal rosette of leaves). In the case of *Datura*, we collected headspace separately from flowers and from foliage because the flower is a valuable nectar source and the leaves are an oviposition substrate for *M. sexta*. Although flowering *Datura* plants receive more eggs than non-flowering *Datura* plants (Reisenman et al. 2010), foliage alone attracts egg-laying females in the field (Allmann et al., 2013) and in the lab (Spaethe et al. 2013). We therefore tested flowers and leaves of *Datura* separately.”

3. How about the male moth's responses to these odors? Generally, both male and female moths search for nectar, while only female moths search for oviposition sites. By analyzing how male and female moths respond to these scents, it may be possible to estimate which scents are associated with the nectar searching or oviposition searching.

It would indeed be interesting to compare results from male and female moths. This is part of future projects in our lab; however, this comparison would go beyond the scope of our present study.

4. In Figure 2A, the GC-EAD recordings of Datura wrightii was not included, why? I think Datura should be one of the most important plants in the system.

We present Agave as an example GC-EAD trace for nectar sources because Agave had the highest number of EAD-active compounds of all headspaces tested (Figure 2B, C). Results for Datura flower and all other headspaces are displayed in the heatmap in Figure 2C, and the EAD-responses of all individual antenna towards all headspaces tested can be found in the source data file S2.

5. Overall, the in vivo calcium imaging experiment is not complete. It is better to link the active glomeruli with some important EAD-active compounds. Are the compounds with high EAD activities the ones that activate strongly the glomeruli?

Our approach was to study activation patterns across the antennal lobe using a puff of plant bouquet. The patterns we observed seemed to be already modulated by presynaptic inhibition via local interneurons (line 397-405). It would indeed be interesting to compare the antennal lobe activity to all individual compounds of a blend with the activity to the whole blend. However, this would need a very tedious concentration control, as all compounds appear at compound-specific concentrations in the blend. Although interesting, this experiment is beyond the scope of the manuscript. However, we discuss an example odorant, which “elicited a strong antennal response, was present in 11 out of 17 plant samples tested, and when tested on its own activates several glomeruli (Bisch-Knaden et al., 2018). However, two of the background plants although releasing this odor, and accordingly evoking a strong antennal response, did not elicit any activity in the antennal lobe” (line 409-413). Repeating the calcium imaging experiments with all individual GC-identified compounds would be an interesting follow-up study but is beyond the scope of the present manuscript. We, therefore, believe that regarding our interest in how full blends of relevant versus irrelevant plants are coded in the brains of virgin and mated females, the calcium imaging experiment is complete.

6. The activation of glomeruli depends on mating status. Is this difference derived from the differential peripheral inputs or the changed modulation on the activity of glomeruli by pre-and postsynaptic modulation?

We did not find an effect of the females’ mating status at the periphery (see original manuscript line 178-180) and can only speculate (and did so) about possible mechanisms for the observed modulation at the level of the antennal lobe (line 478-482).

7. I suggest that the odorants from two hostplants activating the glomeruli in the antenna lobe be identified. It would be very nice if they can also determine the attractiveness of these chemicals to the mated adults.

We agree that identifying those individual components of host plants that might be most responsible for the attraction of mated females is interesting. However, it might as well be that no individual compound can be found because a blend of odorants at specific ratios might be necessary to attract a mated female to a host plant. This topic is part of a project already started in our lab and will be published in future. Our present study, however, focusses on the analysis of spatial activation patterns evoked by the whole, complex plant headspace.

Reviewer #3 (Recommendations for the authors):This is an easy to read, very exciting manuscript. I only have a few minor suggestions for improvement.

We are grateful for this very positive and encouraging review.

In Figure 2C at least in the version I have, some numbers are cropped on the right side (those >10).

We apologize for having accidentally cut the figure at its right margin before pasting it into the text. The missing numbers are 11 (for the occurrence of (Z)-3-hexenyl acetate) and 10 (b-ocimene).

Line 272: graph depicts

We corrected this error.

Line 677: replace "fluorescent" either by "fluorescent dye" or "fluorescence"?

We replaced "fluorescent" by "fluorescent dye".

Discussion lines 310-315:When you discuss the antennal detection of volatiles, you talk about discrimination capability. I would maybe not talk about discrimination at this level. You find indeed differential detection, which could provide them with the necessary information to discriminate, but I would maybe formulate this with a little more caution.In the same paragraph, I am not sure that I can follow your argument that detection and discrimination capability appeared to be better than could be inferred from the chemical analysis.

We rephrased the paragraph in order to clarify our argument and be more careful with our formulation.

“When we tested the antenna of female *M. sexta* with plant headspaces using GC-EAD, we found that the moths in most of the cases detect at least some compounds, even in wind-pollinated background vegetation like grass or careless weed, plants that, based on the chemical analysis, had a weak smell consisting of a small number of components. Female moths, both virgin and mated, therefore seem to be equipped with the sensory capability to not only distinguish between strong and complex scents emitted by nectar sources but also between the bouquets of host plants and surrounding background plants.” (revised line 335-341)

[Editors’ note: what follows is the authors’ response to the second round of review.]

The manuscript has been improved but there are some remaining issues that need to be addressed, as outlined below:The three reviewers are supportive of the revised manuscript and the revisions you made as well as the extensive explanations you provided in the rebuttal. The last suggestions for modification, made by reviewers #1 and #2 involve some explanation to support some statements that will further increase the quality of the manuscript.Reviewer #1 (Recommendations for the authors):I thank the authors for taking the time to carefully consider the reviewers comments, making suggested changes, clarifying some of their statements, etc. The addition of sentences clarifying differences and similarities with previous studies, both methodological and in terms of findings, is very helpful and useful for a reader who is not particularly familiar with the specifics of the system under study.With the changes introduced by the authors, I think that the manuscript now better conveys the novelty and relevance of the findings: while much was previously known about some of the important sources used by hawkmoths for nectar feeding and oviposition (e.g. Datura and Agave flowers, Datura and Prosbocidea foliage), the comparison in terms of chemistry and antennal and antennal lobe responses with those of non-host plant provides a framework for better explaining how moths can distinguish relevant from irrelevant hostplants and nectar sources.I think that in its present form, this is a very interesting study about how specialist insects find olfactory important resources, distinguishing from those that are irrelevant or non-suitable, in their environmentally complex olfactory environments. The combination of field collections in the moths' natural environmental, and of chemical analysis coupled to recording from peripheral olfactory organs and imaging of neuronal activity in the primary olfactory center, plus comparison between virgin and mated females, is a strength of the manuscript.One general comment that I have is regarding the comparison between antennal responses to single (GC-EAD) compounds and antennal lobe responses (vegetative/floral blends), and comparison with previous studies which use GC-multiunit recording. It is not surprising to me that the full blend evokes responses in many glomeruli, as the responses at this level might also be due to interglomerular interactions (reciprocal synapses, inhibition, etc.): it is possible that single compounds evoke responses in few glomeruli, while blends evoke wide-spread responses. I think it is a good addition that the authors now describe the difference between methodologies and what can be learned from each of them. There's also a couple of points (in the list below) that I think the authors should revise (comments on lines 461-471, 502-503, 571-573).Line 20: "Responses to bouquets"

This seems to be a misunderstanding. We refer first to the chemistry of the bouquets, and in the next sentence deal with the responses towards these bouquets (lines 19-23): “Bouquets of larval host plants and most background plants, in contrast, were subtle, thus potentially complicating host identification. However, despite being subtle, antennal responses and brain activation patterns evoked by the smell of larval host plants were clearly different from those evoked by other plants.”

Line 77: the sentence starting with "However…" should be directly after the previous one for better flow of the logic.

Done.

Line 263: "…mating status (Table 2), at least among the 23 glomeruli imaged in this study."

Done.

Line 264: … activated only one glomerulus out of the 23 imaged in the antennal lobe…."

Done.

Regarding Figure 3C: which plant headspace is used here? This needs clarification. I assume is a single species headspace -it wouldn't be appropriate to mix headspaces (the captions says "plant headspaces").

We did not mix headspaces but identified for this analysis the maximum headspace-evoked response for each glomerulus, no matter which headspace did evoke the maximum response. However, *Datura* flower evoked the maximum response in 69% of the cases, and *Agave* flower in 17%.

We added a sentence (underlined) to clarify this.

“Graph depicts for each glomerulus (color code as in A) the average maximum responses (bars) and one standard deviation (whiskers) of 10 virgin and 10 mated females after stimulation with plant headspaces. In 69% of 460 cases (20 maximum values in 23 glomeruli), *Datura* flower was the headspace eliciting the maximum response, and in 17% it was *Agave* flower.”

Line 345: between the bouquets of host plant vegetation and surrounding…"

We rephrased the sentence for clarification (line 343-346).

“Female moths, both virgin and mated, therefore seem to be equipped with the sensory capability to distinguish not only strong and complex scents emitted by nectar sources but also bouquets of host plants and surrounding background vegetation.”

Line 351: "..in typical hawkmoth-pollinated flowers…"

Done.

Line 359: "…. Is at least as sensitive to promising floral blend…"

Done.

Line 367: "…with the duration a female shows feeding behavior (i.e proboscis contact time with a scented filter paper flower, Pearson correlation…., Bisch-Knaden et al. 2018). In contrast…"

We rephrased the sentence (line 367-371).

“EAD responses evoked by these 31 shared odors belonging to seven chemical classes are indeed positively correlated with the duration a female moth shows feeding behavior when encountering the same odors (EAD amplitude *versus* duration of proboscis contacts with a scented filter paper; Pearson correlation coefficient r=0.41, p=0.023).”

Line 371: "… i.e. a behavior related to oviposition….."

Done.

Line 381: "In addition, the antenna might harbor narrowly tuned olfactory receptor neurons strongly responding…"

Done.

Line 414: "… the active GC-peaks overlapped between…"

We rephrased the sentence for clarification (line 410-412).

“In the case of *Proboscidea* headspace, none of its EAD-active compounds found in our experiments was identified in the former study, and vice versa.”

Line 424: "…influence both the composition and the quantity…"

Done.

Line 444: "… presynaptic level, i.e. at the level…"

Done.

Line 450: "…. not elicit activity in the 23 imaged antennal lobe glomeruli."

Done.

Lines 461-471: I think the explanation for the different studies misses the fact that while single components might not evoke broad activity at the AL lobe level, blends/mixtures might do so due to emergent properties of AL circuitry. The current study uses blends as stimuli for imaging of AL activity, which might explain that the authors found broad activation across the array of imaged glomeruli. So the two studies not only used different techniques, each with its own advantages, but seek to answer different questions. Indeed, in their previous publication (Bisch-Knaden et al. 2018), they used monomolecular odorants and for the most part each odorant activates a few glomeruli (at least medium to strong, Figure 2D), including the esters. In line 469 the authors say that ethyl sorbate and benzyl alcohol evoke responses in the AL in the previous study; the responses to ethyl sorbate are small and limited mostly to glomeruli 12 and a few others; similarly, the responses to benzyl alcohol were not very strong and involved about 5 glomeruli. In their previous study, monoterpenes seem to evoke the strongest responses and more widespread (i.e. involving more glomeruli). The way the ms refers to these results gives the reader the impression that the single monomolecular odorants evoke broad responses, comparable to the blend-evoked responses, which I don't think is correct. I suggest that the authors modify this paragraph accordingly.

We added a sentence to mention this further difference between our study and Riffell’s studies (stimulation with a puff of headspace versus a GC-separated stimulus). Line 476-479.

“In addition, we used a puff of the floral headspace as stimulus, i.e. the antennal lobe was activated by the full floral blend, whereas in previous studies a GC-coupled stimulus was applied, i.e. the antennal lobe was activated by temporally separated single compounds present in the floral blend.”

We did not want to raise the impression that single monomolecular odorants in our previous study evoked broad responses comparable to the blend-evoked responses in the present study.

We changed the sentence to clarify this.

“However, compounds that were identified to activate neurons in the male antennal lobe, like ethyl sorbate (Agave) and benzyl alcohol (Datura), were also EAD-active in our present study (Figure 2C) and evoked responses in some glomeruli of the female antennal lobe in a previous imaging study (Bisch-Knaden et al., 2018).”

Line 500: "… to evoke a response in sensilla targeting mostly (but not only) the two female specific glomeruli" (Shields & Hildebrand shows that while most sensilla dye-filled target the LFGs, some target a few other glomeruli).

Shields and Hildebrand cut 5-10 neighboring trichoid sensilla and bathed the respective annulus for 2-3 days in a dye solution. It is not clear if the few not-LFG glomeruli that were stained are really targeted by trichoid sensilla or by other sensillum types that had been accidentally damaged and stained in the same dye pool. Back-fillings from individual trichoid sensilla can answer this question.

We changed the sentence:

“These odors failed to evoke a response in sensilla targeting mainly the two female-specific glomeruli {Shields, 2000}.”

Lines 502-503: I still think that the authors do not have sufficient arguments for the statement at it is in this sentence. This is because: 1) the authors do not find response to vegetation in the LFGs, but the imaging technique, as the authors state, reveals activity from AL afferent mostly, not AL outputs. Although a cultivated plant, King et al. (2000) and Reisenman et al. (2009) reported conspicuous responses to tomato leaves in one of these glomeruli; 2) it is possible that the LFGs act in concert with other glomeruli to guide oviposition behavior (concerted responses not revealed by imaging of afferents might have important downstream effects); 3) males with induced LFGs fly more towards host plants (because these are the most prominent female specific glomeruli, this suggest that these glomeruli process some odorants which directly or indirectly signal an oviposition site. I thus suggest for this line something like this: "In spite of this, it is still possible that these female-specific glomeruli act in concert with other glomeruli to guide the female-specific behavior of identifying an oviposition site, a hypothesis that need further investigation."

We rephrased the paragraph (line 506-519)

“Two enlarged, female-specific glomeruli that are located at the entrance of the antennal nerve into the female antennal lobe — at the same position as the sex pheromone-processing macroglomerular complex in males {Matsumoto, 1981;Rossler, 1998} — seem predisposed to be involved in oviposition choice. This hypothesis is supported by the fact that output neurons targeting both glomeruli respond to headspace of tomato leaves, another host plant for *M. sexta* {King, 2000;Reisenman, 2009}. On the other hand, the two host plant bouquets tested in our imaging experiments did not activate these glomeruli (glomeruli 1 and 2, Table 2), confirming results of a study using vegetative headspace from the hosts *Datura*, *Nicotiana*, and tomato. These scents failed to evoke a response in sensilla targeting mainly the two female-specific glomeruli {Shields, 2000}. Therefore, the question if these glomeruli might be involved in identifying an oviposition site is still open.”

Line 520: "However, host plants activate only one of these glomeruli, …… activated additional glomeruli. While it is possible that host plants activate glomeruli not imaged in this study, the resulting neural representation…"

We rephrased the sentence (line 537-540).

“Even if non-host plants as well as host plants would activate more glomeruli in areas of the antennal lobe that were inaccessible in our imaging study, the resulting neural representation of non-host plants in the antennal lobe of mated females would remain different from the pattern evoked by host plants.”

Line 550: here add Goyret 2010 (J Exp Biol, Look and touch: multimodal sensory control of flower inspection movements in the nocturnal hawkmoth Manduca sexta).

In this paragraph, we discuss the females’ choice of oviposition sites. A citation about cues used during flower inspection movements is not fitting here in our opinion.

Line 552: "…when searching for oviposition…"

Done.

Line 571-573: I still think that the case of *D. wrightii* is particularly interesting and the fact that the plant has a faint vegetative scent but powerful floral odor has significance. It is entirely possible and supported by previous findings that female might simply use the floral odors for long distance olfactory attraction, as these are fragrant and abundant (Raguso et al. 2003, and evoke strong responses). Once in the vicinity or closer, females might use vegetative odors to decided whether or not oviposit on leaf tissues (in addition to feeding on nectar, as it is known that females mix feeding and oviposition bouts). Therefore, the two processes, oviposition and feeding, guided by weak and strong odors but at different timescales, might entangled with each other in the *M. sexta*-*D. wrightii* system. Also, it is commonly observed that *M. sexta* moths mix oviposition and feeding bouts on this plant, and it is reported that flowers increased oviposition both in the lab and in the field (Reisenman et al. 2010). I suggest the authors modify the sentence starting with "Hence, *M. sexta*…" to reflect this fact.

We fully agree with the reviewer’s statements about the long-distance attraction using flower odors, and entangled feeding and oviposition behaviors in the case of *Datura*. However, non-flowering plants receive eggs, and we were therefore interested in a comparison between the odors of *Datura* leaves and *Proboscidea*. To clarify this, we added a sentence at the beginning of the paragraph (line 559-564).

“*M. sexta* females intersperse feeding and oviposition bouts when visiting a flowering *Datura* {Raguso, 2003}, and lay more eggs on flowering than on non-flowering plants {Reisenman, 2010}. However, *Datura* foliage alone attracts egg-laying females in the field (personal observations) and in the lab {Spaethe, 2013;Nataraj, 2021}. We therefore tested leaves of *Datura* separately and compared their headspace with that of *Proboscidea*, the only other host plant in our study area.”

Line 583: "… in contrast to the weak but specific activation of single glomeruli (among those imaged) by host plants of *M. sexta*."

Done.

Reviewer #2 (Recommendations for the authors):The authors responded to the questions I raised, and then I think that this work deserves publication in the journal. However, I still have two suggestions:(1) Since Datura wrightii is a unique and important plant in this study system, which is both the nectar source and host plant of this moth species, I'm sticking to my opinion that the representative traces of GC-EAD and in vivo calcium imaging recordings of headspace stimulations of the flowers and foliage of Datura wrightii should be added in Figure 2A and Figure 3B although these data were reflected in other figures and source data. For floral and foliage odors of Datura could attract females for oviposition, a comparison of responses of antennae or antennal lobes between floral and foliage of the same plant would be significant.

Based on the reviewer’s suggestion, we included representative imaging results for *Datura* flower and *Datura* foliage, both for a virgin and a mated female (Figure 3B), to make a comparison of antennal lobe responses towards the flower and the leaves of *Datura* possible.

(2) The EAD activities and calcium imaging activities are the core contents of this study, it is better to analyze their linkage. As the dye used in vivo calcium imaging experiments is Calcium Green-1 AM, the information reflected in the calcium imaging activity may be mainly the input of the olfactory sensory neurons from the antennae. Therefore, there should be positive relationships between EAD activities and calcium imaging activities in theory. I am very curious about the following questions: based on the authors' previous research (Bisch-Knaden et al., 2018), which compound(s) could elicit activities of the only glomerulus (Glomerulus 4) activated by Datura foliage? Were these compounds present in the headspace blend of Datura foliage? If so, how about EAD responses or even behavioral responses to these compounds? The same questions with Glomerulus 15 and 12 for Proboscidea. These could be discussed in the Discussion section.

We agree that the link between peripheral and central responses is interesting and already discussed this in our original manuscript. Based on the reviewer’s suggestion, we now use *Datura* foliage and its two main active components to illustrate the possible presence of non-linear processing in the moth’s lobe (underlined in the paragraph below, line 425-465), and removed the example we had given in the original manuscript. The observed mixture interactions already at the input level of the antennal lobe preclude simple correlations between EAD and imaging results.

“Bath application of a fluorescent calcium-sensor allows monitoring of odor-induced neural activity in the brain. Each neuron type in the treated brain region might take up the marker molecules. However, as each glomerulus in the antennal lobe receives input from 4000-5000 olfactory sensory neurons {Oland, 1988}, and is targeted by only four to five projection, i.e. output neurons {Homberg, 1988}, odor-evoked activation patterns in calcium imaging experiments can be assumed to reflect mainly the activity of input neurons. Additionally, about 360 local interneurons per antennal lobe {Homberg, 1988} with inhibitory and/or excitatory functions {Reisenman, 2011} might synapse back onto the sensory neurons, thus modulating their activity and accordingly the observed calcium signal. Although most of these interneurons arborize in many, if not all glomeruli, some interneurons have a more restricted innervation pattern and connect only a few glomeruli {Christensen, 1993}. This type of interneuron seems predisposed to play a role in the coding of complex odor blends released by plants. Interestingly, patchy interneurons are present mainly in female *M. sexta* {Matsumoto, 1981}. In the vinegar fly *Drosophila melanogaster*, patchy interneurons are responsible for non-linear processing of binary odor mixtures {Mohamed, 2019}. For some glomeruli in *D. melanogaster*, this modulation occurred already at the presynaptic level, i.e. at the level we monitored in our calcium imaging experiments. To estimate if non-linear interactions might occur in the antennal lobe of *M. sexta*, we compared headspace-evoked activation patterns with activation patterns evoked by EAD-active, single compounds that were present in the respective headspace. From the bouquet of *Datura* foliage, for example, (Z)-3-hexenyl acetate elicited the strongest antennal response (Figure 2C) and activates mainly four glomeruli (glomeruli 6, 13, 16, and 12) when tested on its own {Bisch-Knaden et al. 2018}. After stimulation with the complex headspace, however, none of these glomeruli was responding (Figure 3E, Table 2). The second best antennal activator in the headspace of *Datura* foliage, geraniol, activates mainly three glomeruli (glomeruli 6, 4, and 5, {Bisch-Knaden, 2018}). Of these, glomerulus 4 was the only one responding towards stimulation with the headspace (Figure 3E, Table 2). We thus conclude that there are indications of local inhibition, as we otherwise would have expected to observe more activated glomeruli after stimulation with the complex headspace. A similar inhibition of glomeruli in mixtures of odors was reported in a calcium imaging study in honeybees, where the inhibitory effect was stronger in ternary than in binary mixtures {Joerges, 1997}. As the plant bouquets tested in our study contained up to 20 EAD-active components, and as local interneurons in *M. sexta*, like in most insects, are mainly inhibitory {Christensen, 1993}, the observed inhibitory mixture interactions after stimulation with complex blends seem plausible.”

In the case of *Proboscidea*, β-ocimene was the best antennal activator. When tested on its own, β-ocimene activates mainly glomeruli 6, 12, and 17 {Bisch-Knaden et al. 2018}. However, headspace of *Proboscidea* activated only glomerulus 15 in virgin females (in our previous study we investigated only virgin females), which again indicates mixture interactions already at the input level of the antennal lobe.